# Towards Unifying Interpretability and Control: Evaluation via Intervention

## Abstract

With the growing complexity and capability of large language models (LLMs), a need to understand model reasoning has emerged, often motivated by an underlying goal of controlling and aligning models. While numerous interpretability and steering methods have been proposed as solutions, they are typically designed either for understanding or for control, seldom addressing both, with the connection between interpretation and control more broadly remaining tenuous. Additionally, the lack of standardized applications, motivations, and evaluation metrics makes it difficult to assess these methods' practical utility and efficacy. To address the aforementioned issues, we propose intervention as a fundamental goal of interpretability and introduce success criteria to evaluate how well methods are able to control model behavior through interventions. We unify and extend four popular interpretability methods—sparse autoencoders, logit lens, tuned lens, and probing—into an abstract encoder-decoder framework. This framework maps intermediate latent representations to human-interpretable feature spaces, enabling interventions on these interpretable features, which can then be mapped back to latent representations to control model outputs. We introduce two new evaluation metrics: intervention success rate and the coherence-intervention tradeoff, designed to measure the accuracy of explanations and their utility in controlling model behavior. Our findings reveal that (1) although current methods allow for intervention, they are inconsistent across various models and features, (2) lens-based methods outperform others in achieving simple, concrete interventions, and (3) interventions often compromise model performance and coherence, underperforming simpler alternatives, such as prompting, for steering model behavior and highlighting a critical shortcoming of current interpretability approaches in real-world applications requiring control. Code is made available for replicability.

## 1 Introduction

As large language models (LLMs) have become more capable and complex, there has emerged a need to better understand and control these models to ensure their outputs are safe and human-aligned. Many interpretability methods aim to address this problem by analyzing model representations, attempting to understand their underlying computational and reasoning processes in order to ultimately control model behaviour. While many of these methods, and interpretability as a field more broadly, claim control and intervention as abstract goals and present compelling qualitative results demonstrating that intervention is possible (for example, consider Anthropic's Golden Gate Claude Anthropic; Templeton et al. (2024)), the link between interpretation and intervention is often tenuous in practice, as many methods are not explicitly tailored for both. Furthermore, even fewer are thoroughly and systematically evaluated for the ability to control model outputs beyond simple qualitative examples. We believe the reason for this is threefold. First, interpretability methods produce explanations in disparate feature spaces, such as the token vocabulary, probe predictions, or learned auto-interpreted features, hindering comparisons across methods. Second, there exists a "predict/control discrepancy" Wattenberg & Viégas (2024), where the features identified by interpretability methods for *predicting* behavior are not the same as those used for *steering* it. Third, there do not exist standard systematic benchmarks to measure intervention success.

In this work, we view intervention as a fundamental goal of interpretability, and propose to measure the correctness of interpretability methods by their ability to successfully edit model behaviour. In particular, we focus on sparse autoencoders, Logit Lens, Tuned Lens, and linear probing, and

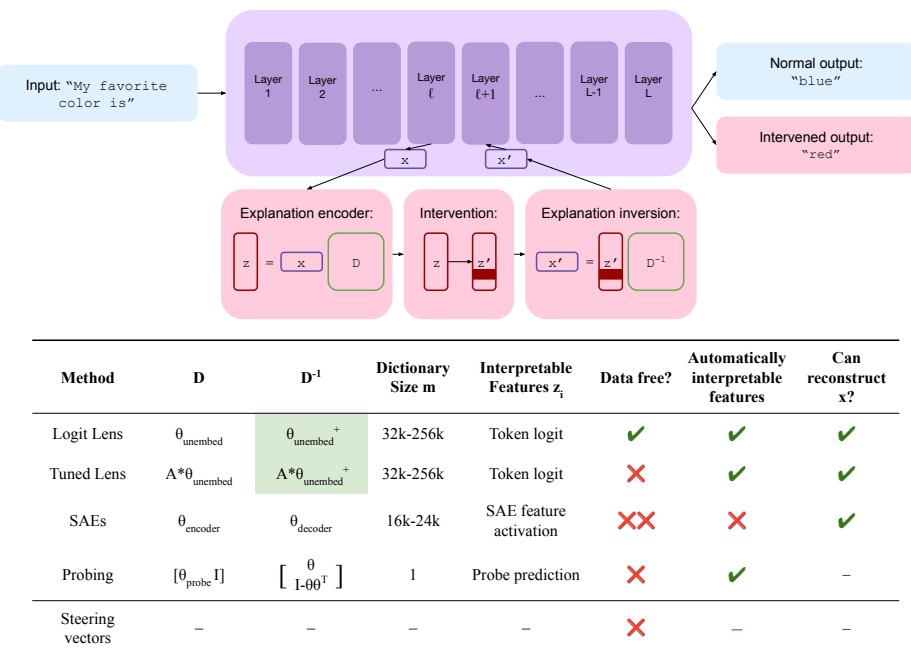

Figure 1: Our proposed intervention framework, which encodes model latent representations, $x$, into human-interpretable features, $z = xD$, that can then be perturbed to $z'$ and decoded back into counterfactual latent representations, $\hat{x}'$. Note that the two green squares are our contribution.

benchmark them with steering vectors and prompting as baselines for intervention. In order to enable comparison across these various methods, we first unify and extend the methods as instances of an abstract encoder-decoder framework, where each method encodes uninterpretable latent representations of language models into human-interpretable features and the decoder of the framework inverts this mapping, allowing us to reconstruct a latent representation from the features. Under this abstract framework, we can intervene on the interpretable feature activations generated by each method and decode them into latent counterfactuals, which produce counterfactual outputs corresponding to the desired intervention.

The unifying feature interpretation and intervention framework allows us to propose two standard metrics for evaluating mechanistic interpretability methods: (1) intervention success rate, which measures how well intervening on an interpretable feature causally results in the desired behavior in the model outputs, and (2) coherence-intervention tradeoff, which measures how well the causal interventions succeed without damaging the coherence of the model's outputs. We evaluate Logit Lens, Tuned Lens, sparse autoencoders, and linear probes, for these metrics on GPT2-sm, Gemma2-2b, and Llama2-7b, comparing them to simpler but uninterpretable baselines of steering vectors and prompting. Our results show that while existing methods allow for intervention, they are inconsistent across features and models. Furthermore, lens-based methods outperform all other methods including sparse autoencoders for simple, concrete features. This is likely due to frequent mislabeling of the learned SAE features, where a feature labeled "references to coffee" actually seems to encode for 'beans' and 'coffee beans' more than 'coffee'. Similarly, probes and steering vectors are prone to learning spurious correlations in their training data, whereas Logit Lens and Tuned lens do not suffer from the same sensitivities. Finally, we show that intervention often comes at the cost of model output coherence, underperforming simple prompting baselines, presenting a critical shortcoming of existing methods in real-world applications that require control and intervention. We conclude this work with detailed takeaways about the strengths and weaknesses of each method, including discussion of which methods are optimal for specific intervention topics, which are best to use out of the box, and which hold the most promise for future development.

Our main contributions include:

- In Section 3.1, we present an encoder-decoder framework that unifies four popular mechanistic interpretability methods: sparse autoencoders, logit lens, tuned lens, and probing.

- In Section 3.2, we propose two evaluation metrics for encoder-decoder interpretability methods, namely (1) intervention success rate; and (2) the coherence-intervention trade-off to evaluate the ability of interpretability methods to control model behavior. We also provide a dataset for benchmarking state-of-the-art LLMs on these metrics.

- In Section 4, we perform experimental analysis on GPT-2, Gemma2-2b, and Llama2-7b, and present detailed takeaways comparing interpretability- and control-based methods.

We hope that this paper encourages future work in the interpretability community to make explicit the intent and application of methods, particularly in reference to control and intervention, and to evaluate and benchmark causality and utility in application.

## 2 RELATED WORK

**Mechanistic Interpretability.** Existing work in mechanistic interpretability broadly falls into two categories: activation patching and interpreting hidden representations. Activation patching utilizes carefully constructed counterfactual representations to study which neurons or activations play key roles in model computation, ideally localizing specific information to individual layers, token positions, and paths in the model Geiger et al. (2021); Vig et al. (2020). However, recent work points to key limitations of patching, particularly with respect to real-world utility in downstream applications such as model editing Hase et al. (2024); Zhang & Nanda (2023). As such, we focus primarily on methods for inspecting hidden representations, of which probes are the most commonly used Alain (2016); Belinkov & Glass (2019); Belinkov (2022). Other methods such as nostalgebraist; Dar et al. (2023) project intermediate representation into the token vocabulary space, with Belrose et al. (2023); Din et al. (2023); Geva et al. (2022) building and improving upon these early-decoding strategies. Ghandeharioun et al. (2024a) unifies most of these methods into an abstracted framework for inspecting model computation. More recently, sparse autoencoders and dictionary learning have been explored as a solution to the uninterpretability of model neurons, particularly due to issues with polysemanticity and superposition Elhage et al.; Bricken et al. (2023); Cunningham et al. (2023); Bhalla et al. (2024); Gao et al. (2024); Rajamanoharan et al. (2024a); Templeton et al. (2024).

**Evaluation.** Due to the recency of the field, standard evaluation metrics across interpretability methods have not yet been established, and similar to the broader interpretability field, evaluation is frequently ad-hoc and primarily qualitative in nature. With regards to quantitative metrics, in Gao et al. (2024); Templeton et al. (2024); Makelov et al. (2024), sparse autoencoders are evaluated for reconstruction error, recovery of supervised or known features, activation precision, and the effects of ablation; however, none of these metrics measure the human interpretability or correctness of explanations. Sparse autoencoder evaluation has also sometimes included ratings by humans to determine the interpretability of the learned features for labeling, as done in Rajamanoharan et al. (2024b), but this is completely unrelated to evaluations of correctness or utility.

**Causal Intervention.** Previous literature on probing frequently evaluates learned probes and features through intervention to ensure causality and correctness, as done in Li et al. (2022); Chen et al. (2024); Hernandez et al. (2023b;a). The interventions performed for measuring causality are similar to those used to perform model "steering" Rimsky et al. (2023); Panickssery et al. (2024); Ghandeharioun et al. (2024b) and should ideally produce the same effect but with the added claim of interpretability. Beyond probing, only Belrose et al. (2023); Chan et al. (2022); Olah et al. (2020); Templeton et al. (2024) consider causal intervention as a tool for evaluation. However, Belrose et al. (2023); Olah et al. (2020); Chan et al. (2022) mainly focus on mean ablation for intervention, with no exploration of the coherence of the counterfactual outputs or their utility in application. Templeton et al. (2024) on the other hand only provides a qualitative evaluation of intervention via their 'Golden Gate Claude' [Anthropic] and does not compare to other interpretability methods.

## 3 METHOD

In this section, we first introduce a unifying framework for four common mechanistic interpretability methods: sparse autoencoders, Logit Lens, Tuned Lens, and probing, along with modifications to these methods that permit principled intervention on representations. We then propose evaluation

metrics for (1) testing the correctness of explanations via intervention and (2) the usefulness of these methods for steering and editing representations and model outputs.

### 3.1 UNIFYING INTERVENTION FRAMEWORK

**Latent vectors to interpretable features.** The central aspect of most interpretability work is the ability to translate model computation into human-interpretable features, whether the computation be latent directions, neurons, components, reasoning processes, etc. Many works aiming to explain LLMs focus particularly on hidden representations, where the mapping between high-dimensional dense embeddings and human-interpretable features is modeled through a (mostly) linear dictionary projection or affine function:

$$z = f(x) = \sigma(x \cdot D)$$
$$\hat{x} = g(z) \approx f^{-1}(z) = z \cdot D^{-1}$$
$$z' = \text{Edit}(z), \;\; \hat{x}' = g(z')$$

where each $z_i$ is a feature activation, each $i$ in $D$ corresponds to a human-interpretable feature[1], and $\sigma$ is an activation function that is frequently the identity. In the case of **sparse autoencoders**, $D$ is a learned, overcomplete dictionary, with $N$ in range 16k - 65k for small models (up to 16M for large models), and $\sigma$ is a ReLU, JumpReLU, or ReLU + top-k activation function. Given that SAE features are learned, they are not immediately interpretable and must be labelled by humans or strong LLMs. For **Logit Lens**, $D$ is simply the language model's unembedding matrix, meaning each feature corresponds to a single token in the vocabulary. For **Tuned Lens**, $D$ is the exact same as Logit Lens but with a learned linear transformation applied. **Linear probes** can be thought of as a learned dictionary with $N = 1$ where $\sigma$ is a sigmoid or softmax activation. Of all these methods, Logit Lens is the only method that does not require any training data, and sparse autoencoders are the only method that does not produce immediately interpretable features. Please see Figure 1 for a visual summary of these methods.

**Interpretable features to counterfactual latent vectors.** While producing explanations is straightforward for each of these methods, intervening on model representations using the information provided by explanations is not as simple. Doing so requires defining a reverse mapping from the explanations to the latent representations of the model, which is only explicitly done by sparse autoencoders. We defined the inverse mappings for the rest of the functions as follows. To map **Logit Lens**'s explanations back into the model's latent space, we would ideally apply the inverse of the unembedding matrix to $z$; however, in practice this is often ill-conditioned due to the dimensionality of $D$. As such, we instead use the low-rank pseudoinverse of the unembedding matrix and right-multiply it to the explanation logits. Similarly, for **Tuned Lens**, we model the decoding process through the pseudo-inverse of the Tuned Lens projection applied to the unembedding matrix. Notably, both of these methods only require a simple linear transformation to go back-and-forth between latent vectors and explanations. For **probing**, $D^{-1}$ can be defined such that $\hat{x}' = x + \theta$, as proposed by Chen et al. (2024) which adds the linear probe's weights to the latent representation, bypassing the need for optimization. **Sparse autoencoders** have a well-defined backwards mapping through the SAE decoder, which is frequently linear in practice and often the transpose of the encoder weights. Thus, we are now able to decode $z$ into $\hat{x}$ for all methods.

**Intervening on interpretable features.** Given the above framework, intervention is performed by directly altering the feature activation $z_i$ corresponding to the desired feature $i$ to be edited. While the edited activation $z_i'$ can naively be set to some constant value $\alpha$, the same constant may have drastically varying effects for different tokens and different prompts. As such, to take into account the context of $z$, for Logit Lens, Tuned Lens, and SAEs we set $z_i' = \alpha * \max(z)$. This ensures that the feature $i$ is the most dominant feature in the latent vector for $\alpha > 1$. Decoding $z'$ yields the altered latent representation $\hat{x}' = g(z')$, which accounts for both the error of the explanation method as well as the intervention performed. For probing and steering vectors, $\hat{x}' = x + \alpha * v$,

---

[1]While "features" are not necessarily well-defined in the interpretability community, we find that the second definition in Elhage et al. most closely aligns with our own notion of features as human-interpretable concepts that are relevant for manipulation and editing.

where $v$ is the steering vector or the weights of the linear probe. Note that $\alpha$ is a hyperparameter that must be tuned for each method, model, and sometimes even intervention feature and thus cannot be used to compare the effects of interventions across methods. In order to do so, we can instead measure the normalized difference between the latent vectors $x$ and $\hat{x}'$, to better understand the relationship between the degree of intervention performed and our various evaluation metrics. We also note that $\hat{x}$ and $\hat{x}'$ are not necessarily in-distribution for the language model, but due to the additive nature of the residual stream, the linear representation hypothesis, and the linear nature of the decoder/unembedding matrix, we believe that such interventions may still be principled in practice (see Park et al. (2023) for more on the linear representation hypothesis and intervention).

**Open-ended generation.** In order to generate open-ended text after intervening on the explanation, we edit the corresponding representations *in place*, as is common practice with prior steering methods. Formally, the representation $x_t$ at token position $t$ and layer $l$ is edited to be $\hat{x}_t'$, ensuring a causal effect on all ensuing tokens $x_{t+1}, x_{t+2}, ..., x_T$.

## 3.2 Comparative Evaluation Across Methods and Models

Given the overall lack of standardized evaluation of mechanistic interpretability methods, we intend for this work to serve as a starting point for systematic evaluation by evaluating methods in simple, easy-to-measure contexts. In particular, we think of our evaluations as measuring a kind of upper bound for these methods: in the easiest of settings, how well do existing methods work?

**Explanation Correctness** We first propose metrics to evaluate the *correctness* of explanations and interventions. More specifically, to test whether a single feature of an explanation $z_i$ is correct, we intervene on that feature to produce $z_i'$ and decode $z'$ to $\hat{x}'$. We can then hypothesize about the corresponding change in the model's output when generating from $\hat{x}'$ beforehand, as it should match the intervention made to produce $z'$. For example, if feature $i$ encodes the concept "references to Paris," increasing the value of $z_i$ should result in increases to references of Paris in the model's output. From this, we propose a metric of **Intervention Success Rate**, which measures if increasing the activation of feature $i$ by increasing $z_i$ results in the appropriate increase of the feature $i$ in the model's output. To evaluate a continuous relaxation of this, we can also similarly measure the probability assigned to tokens relating to feature $i$. As such, even if the model's output does not directly reflect interventions made to $z_i'$ due to sampling, we can measure if increasing the activation of $i$ results in any change to the model's output at all. We refer to this metric as **Intervened Token Probability**. Importantly, both of these metrics can be thought of as measuring the causal fidelity of the features highlighted by explanations.

**Usefulness of Intervention Methods** While intervention is a useful method for evaluating the correctness of explanations, it is also a desideratum of its own and a frequent motivation for many explanation methods. For example, methods are often developed for the purpose of de-biasing model outputs or increasing model safety, either by localizing bad behavior or identifying it at inference time, thus allowing for targeted edits to be made. However, a lack of this direct connection between interpretation and model intervention has led to illusory results in prior literature, similar to predict/control discrepancies Hase et al. (2024); Wattenberg & Viégas (2024). By directly and explicitly measuring how effective interpretability methods are at allowing for targeted intervention or steering, we can avoid such failure cases. Importantly, intervention is only useful if the language model retains its overall performance and the text output by the model after the intervention still satisfies the purpose of the query as well as the intervention. In particular, we are interested in whether interpretability methods can be used to steer model outputs towards feature $i$ without damaging the model's purpose as an LLM. We do so by measuring the **Coherence** of the model's outputs after the intervention has been applied. This can be done by any oracle, such as a human or another capable LLM. Similarly, we can also measure the **Perplexity** of the intervened outputs with respect to a strong language model. In practice, we use Llama3.1-8b for both of these metrics, as it is reasonable sized, high-performing, and open source, allowing for the measurement of perplexity. To measure coherence, we use the prompt "*Score the following dialogue response on a scale from 1 to 10 for just grammar and comprehension, ignoring incomplete sentences.* \n\n *Context:* {`Prompt`}\n *Dialogue Response:* {`Output`}\n\n *Score* =".

**An Open-ended Evaluation Dataset.** In order to evaluate these methods to the best of their capabilities, we are interested in assessing their ability to intervene when intervention is straightforward

Table 1: Normalized latent reconstruction error without intervention.

| Method | Gemma2-2b | Llama2-7b | GPT2-sm |
|--------|-----------|-----------|---------|
| Logit Lens | 0.52 | $5e^{-5}$ | 0.22 |
| Tuned Lens | – | $5e^{-3}$ | 0.32 |
| SAEs | 0.38 | – | 1.64 |

and possible given the prompt. Consider the question "*What is $\iint sin(3*x)*cos(y)dxdy$?*". Intervening on the model's output for this prompt with a feature related to unicorns is not necessarily intuitive, as there is a correct answer to the prompt that is entirely unrelated to the intervention topic. As such, we want to evaluate these methods on prompts that allow for steering towards a variety of topics or features. To that end, we construct a dataset of 210 prompts related to poetry, travel, nature, journaling prompts, science, the arts, and miscellaneous questions that could plausibly be answered while satisfying a variety of intervention topics. All prompts are open ended to allow for many potential answers. Some example prompts include "*In ten years, I hope to have accomplished*", "*Something cool I learned about from a Biology Ph.D. student was*", and "*What is your favorite dad joke?*". This dataset is released as part of the accompanying GitHub repository.

## 4 EXPERIMENTS

In this section, we evaluate the five methods on Gemma2-2b Team et al. (2024), Llama2-7b Touvron et al. (2023), and GPT2-sm Radford et al. (2019). We first measure the completeness of the explanations produced in Section 4.2. We then benchmark the methods on the metrics described in Section 3.2, providing quantitative results in Sections 4.3 and 4.4 and qualitative results in Section 5. Finally, we present an analysis of the empirical directional similarities between methods in Section 4.5, and analyses of the efficacy of methods at different model depths in Appendix A.6.

### 4.1 IMPLEMENTATION DETAILS

**Intervention Topics.** We pick 10 intervention topics that all relate to references to specific words or phrases: {'beauty', 'chess', 'coffee', 'dogs', 'football', 'New York', 'pink', 'San Francisco', 'snow', 'yoga'}. These simple, low-level features are ideal for evaluation through intervention for four key reasons: first, measuring the presence of a word or phrase is much easier than measuring a high-level abstract concept such as sycophancy, second, these features were available across many layers for the Gemma2-2b and GPT2-sm sparse autoencoders, third, the features necessarily exist in the Logit Lens unembedding dictionary, and finally, datasets that are labelled for the presence of these features are very straightforward to collect for generating steering vectors and probes. As such, we can easily compare intervention on these features across methods and also easily measure intervention success by checking if the given word/phrase exists in the model's output.

**Steering vectors and probing.** We implement steering vectors as described in Rimsky et al. (2023) with a few simple modifications. Where in Rimsky et al. (2023), the difference between contrastive pairs is taken only at the last token, we find that averaging across the token dimension and taking the difference between those averages yields much better results. This is due to the fact that in Rimsky et al. (2023), the only difference between representations occurred in the token position of the answer letter, or the last token; however, in our case the information related to the intervention feature could be present at any token. Contrastive data pairs were generated by prompting ChatGPT for 200 pairs sentences or phrases about the desired intervention topic and about other topics within the same genre. For example, we used the prompts "Can you please generate 100 short sentences or phrases with the color 'pink' in them?" to create positive samples and "Can you also generate 100 phrases or sentences with other colors (not pink) in them?" to create negative ones. We also use these contrastive pairs to train the linear probes, using the implementation from Chen et al. (2024). All probes reached train and test accuracies of 100% across all models and intervention topics.

**Sparse autoencoders.** We focus specifically on sparse autoencoders trained to interpret the residual stream of transformer models. We use the SAELens library from Bloom (2024b) for GPT2-small and the Gemma Scope SAEs released by Google in Lieberum et al. (2024) for Gemma2-2b. For

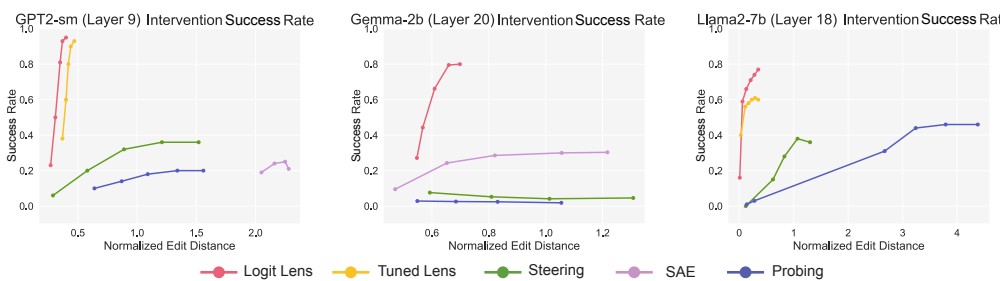

Figure 2: Evaluation of the Intervention Success Rate for each method. Note that normalized edit distance is a proxy for intervention intensity that is comparable across methods. Logit Lens outperforms all other methods across all models.

both models, SAE feature labels were found via Neuronpedia Lin & Bloom (2023), which allows users to interact with, explore, and search for auto-interpretation labelled learned features. Note that the Gemma SAEs are of the JumpReLU architecture from Rajamanoharan et al. (2024b) whereas the GPT2-small SAEs are standard ReLU + L1 regularization SAEs, as noted in Bloom (2024a).

## 4.2 SANITY CHECKING EXPLANATION RECONSTRUCTIONS

Before testing these methods for their ability to intervene, we first want to evaluate the completeness of the explanations and the effect of replacing $x$ with $\hat{x}$ *without* any intervention or editing. We do so by measuring the normalized latent reconstruction error: $\texttt{Error} = ||\hat{x} - x||/||x||$ where $\hat{x} = g(f(x)) = g(z)$. This error is a key part of the loss function that sparse autoencoders are trained on and measures the information loss incurred by mapping between the language model's latent space and the interpretable feature space. Given that steering vectors and linear probes do not output complete explanations, we only measure this error for the other three methods, as shown in Table 1, where we see that errors vary a lot across models but most methods are relatively consistent in their error, with the exception of the GPT2-sm sparse autoencoders.

We also measure the coherence of the outputs produced by replacing $x$ with $\hat{x}$, as shown in Appendix Figure 7. We find that the coherence of the outputs generated by the reconstructed latents generally matches the coherence of the clean model outputs. We use a deviation of $\pm 1$ around the mean of clean output coherence scores as a threshold for future evaluations, shown in the dashed lines.

## 4.3 INTERVENTION SUCCESS ACROSS MODELS

As described in Section 3.2, in order to evaluate the correctness of explanations, we measure the causal effects of intervening on specific features of each explanation. For a given feature or intervention topic $i$, we see if increasing the activation of that feature results in an increase of the feature in the model's output in the top row of Figure 2. In order to compare across methods, which all have different explanation feature spaces and scales, we measure the success of interventions as a function of the norm of the distance between the edited latent representation $\hat{x}' = g(Edit(f(x)))$ and the original latent representation $x$: $||\hat{x}' - x||/||x||$ as done in Section 4.2. Results for intervened token probability are in Appendix A.4.

We find that across all three models, Logit lens and Tuned lens have the highest intervention success rate, regardless of the normalized edit distance. While increasing intervention strength and editing the latent representations more for steering vectors and probes results in greater intervention success for GPT2-sm and LLama2-7b, the same is not true for Gemma2-2b, for which intervening with steering vectors and probes seems to have minimal effect. Furthermore, we find that intervening with sparse autoencoders is moderately successful, but due to the large reconstruction error for GPT2-sm, even with intervention, the edit distance is much greater than that needed for Logit lens and Tuned lens. In general, we believe the lower performance of Sparse Autoencoders is due to heavy noise in the labels of features. For example, a feature labelled 'references to coffee', is sometimes actually a feature that encodes for references to 'beans' and 'coffee beans', and thus only sometimes increases mentions of 'coffee'. Probes and steering vectors also have suboptimal performance, often due to learning of spurious correlations in the training data rather than the true intervention feature.

When measuring token probability rather than output presence, as explored in Appendix A.4, we see that intervention with all methods across all models increases the probability of intervention-related tokens, even if the intervention does not succeed. We also note that there is a significant difference between the order of magnitude of the intervened token probability for sparse autoencoders, around $10e^{-5}$ and the rest of the methods, which range from $10e^{-4}$ to 0.5.

## 4.4 EFFECTS OF INTERVENTION ON OUTPUT QUALITY

We next measure the coherence of the intervened output text produced by each method to ensure that intervention through mechanistic interpretability methods is possible in applications and does not damage the utility of the model. We measure coherence as described in Section 3 as a function of the normalized latent edit distance $||\hat{x}' - x||/||x||$ and with respect to the intervention success rate in Figure 4. We visualize the mean of coherence scores for the clean model outputs with solid black horizontal lines, the same as those shown in Figure 7, with a buffer of $\pm 1$ around the mean in dashed lines. We also consider a prompting baseline, where we simply prompt the language model to constantly talk about the intervention topic, to better understand the optimal coherence possible while satisfying the intervention. This is shown by the teal stars in 4. Given that only Llama2-7b was instruction-tuned out of the models evaluated, prompting was less successful for Gemma2-2b and infeasible for GPT2-sm. Also note that the intervention success rate approaches near 100% with prompting as the number of generated tokens increases; however, seeing as we only generate 30 tokens, the success rate is slightly lower than expected.

We find that for GPT2-sm and Llama2-7b, intervention with most methods results in a sharp decline in text coherence, with Logit Lens and Tuned Lens having the most dramatic fall-off. In fact, we find that even with a full point drop in coherence from the baseline models' coherence (denoted with the black lines), the representations can barely be edited by any methods except for Gemma2-2b. While at first, this seems quite concerning, as this would prevent the real-world application of these methods to scenarios necessitating high model performance, viewing coherence as a function of intervention success rate tells a different story, as shown in Figure 4. In fact, we see that Logit Lens and Tuned Lens have relatively good coherence-intervention tradeoffs for GPT2-sm and LLama2-7b, with success rates of around 0.5 and 0.6 respectively when considering outputs within one point of deviation from the mean coherence score of the clean model. All other methods exhibit much less desirable Pareto curves. We also note that as the models increase in size and performance, with GPT2-sm being the smallest and worst language model and Llama2-7b being the largest and most performant, the Pareto curves are shifted up and to the right more for most methods, suggesting that targeted intervention is more possible as models become more capable.

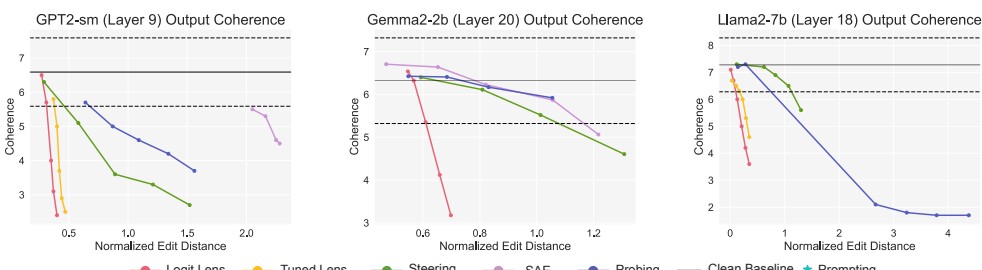

Figure 3: Intervened model output coherence measured with respect to normalized edit distance.

**Qualitative Examples.** In addition to our empirical results, we present examples of the output text for all intervention methods in Figure 5 for qualitative evaluation of intervention on the feature 'yoga'. Examples for the "Optimal intervention strength" (left column) were randomly chosen from the outputs where intervention succeeded and coherence was still relatively high. Examples for "Excessive Intervention" were randomly chosen from the outputs of the highest intervention strength tested (right column). Note that intervention results in infinite repetition at very high intervention strengths for all methods; however, only Logit Lens and Tuned Lens result in repetition of tokens related to 'yoga'. More examples are provided in Appendix A.7.

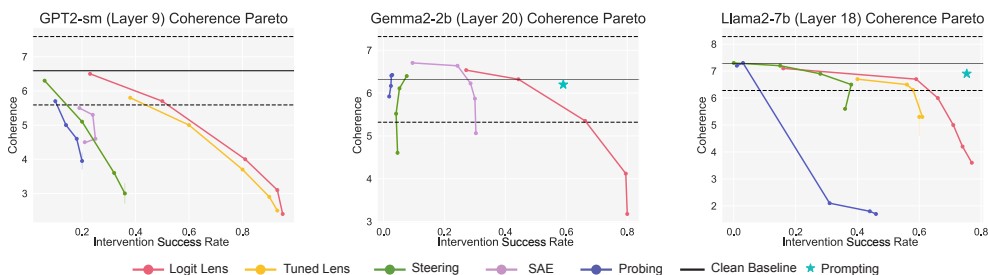

Figure 4: Intervened output coherence measured with respect to intervention success rate.

| Method | Optimal Intervention Strength | Excessive Intervention |
|---|---|---|
| **Logit** | On my next vacation, I hope to visit the beautiful country of Costa Rica. I've been reading a lot about the country and I'm really excited about the **yoga** retreats I' | My favorite song from the 21st century is "**Yoga**, **Yoga**, **Yoga**" by **Yoga Yoga Yoga Yoga Yoga Yoga Yoga Yoga Yoga Yoga** |
| | (Edit Distance: 0.6, Coherence: 7, Model: Gemma2) | (Edit Distance: 0.7, Coherence: 3, Model: Gemma2) |
| **Probing** | Think about a time you felt truly at peace. Maybe you were sitting on a beach, or maybe you were sitting in a **yoga** studio. Maybe you were sitting in your own home, or maybe you | My favorite fiction author is the one and only, the one and only, the one and only, the one and only, the one and only, the one and only, the one and only, |
| | (Edit Distance: 0.7, Coherence: 5, Model: Gemma2) | (Edit Distance: 1.1, Coherence: 4, Model: Gemma2) |
| **SAE** | The coolest wildlife I've ever spotted was a **yoga** class in a tree. I was in the jungle of Costa Rica, and I was doing a **yoga** class in a tree. I was in the jungle of Costa Rica, and I was doing a **yoga** | My favorite museum to visit is the Hat **yoga** practice sequence. It is a sequence that is a sequence that is a sequence that is a sequence that is a sequence that |
| | (Edit Distance: 0.9, Coherence: 6, Model: Gemma2) | (Edit Distance: 1.2, Coherence: 5, Model: Gemma2) |
| **Steering** | My favorite scientist is Dr. Deepak Chopra. I have been practicing **yoga** for 7 years and I have been practicing meditation for 4 years. I have been practicing **yoga** | Last year, I went to a practice session for a beginner class I was initially initially initially initially initially initially initially initially initially initially |
| | (Edit Distance: 0.6, Coherence: 5, Model: Gemma2) | (Edit Distance: 1.3, Coherence: 5, Model: Gemma2) |
| **Tuned** | My favorite thing to do outside is to go to the gym. I'm not a gym rat, but I'm a **yoga** instructor. I'm a **yoga** teacher. I'm a **yoga** | An artist whose work I admire is **Yoga**ogaogaogaogaogaogaogaogaogaogaogaogaogaogaogaogaogaogaogaogaogaogaoga |
| | (Edit Distance: 0.4, Coherence: 5, Model: Llama2) | (Edit Distance: 0.5, Coherence: 2, Model: Llama2) |

Figure 5: Examples of intervened model outputs for intervention feature '**yoga**' at both the optimal intervention strength (left) and the maximum intervention strength tested (left). Outputs degrade into incoherent repetition at high intervention strength for all methods.

## 4.5 INTERVENTION SIMILARITY BETWEEN METHODS

We finally evaluate the empirical similarity between each of these methods. By converting each intervention into one in the latent space of the model, we can measure the cosine similarity between *edit directions* for each method. More specifically, for each method and intervention topic, we compute the expectation of $\hat{x}' - x$ over all samples. This difference vector can be thought of as the direction of the intervention topic in the model's latent space. We compute the cosine similarity between these vectors for each intervention topic and display the average across topics in Figure 6.

We find that Logit Lens and Tuned Lens are highly similar, as expected. Similarly, steering vectors and probe weights tend to lie in similar directions, likely due to the same underlying data used to train both. Most interestingly, we find that sparse autoencoders tend to intervene in somewhat similar directions to steering vectors and probes. Even more surprisingly, they have near orthogonal directions to Logit Lens and Tuned Lens, even when interventions succeed for all methods. This is likely due to the high-dimensionality of model latent spaces, allowing for many vectors to be near-orthogonal. We speculate that sparse autoencoders may be more similar to probes and steering vectors because the three methods may have a bias toward representing past information and tokens, due to their training and labelling algorithms. Logit Lens and Tuned lens, on the other hand, are designed to reveal information about the *next token* specifically, given that they are early-decoding strategies and thus may contain more information about model outputs rather than inputs.

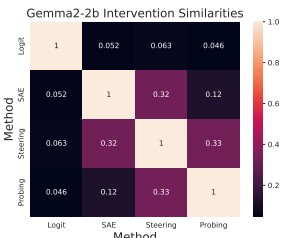 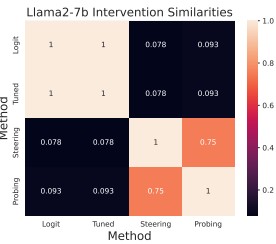

Figure 6: Cosine similarity of intervention directions in model latent space across methods. Logit Lens and Tuned Lens are highly similar, as well as steering vectors and probes.

## 4.6 HIGH-LEVEL COMPARISON OF METHODS

Given the differences between methods, including how their features are learnt, what explanations they produce, and the causal fidelity and utility of explanations for steering, we present the following characterization of the strengths and weaknesses of each method.

**Logit Lens** is easy to use, requires no training, and maps features directly to vocabulary tokens, making it highly interpretable, and we find that it generally has high causal fidelity as well. However, its predefined, static features are limited and rudimentary, preventing its utility for steering in many real-world or safety-critical applications where abstract features such as truthfulness and bias are more relevant.**Tuned Lens** shares these traits but requires an additional learned affine transformation, although many are already open-sourced. **Sparse autoencoders** learn a very large dictionary, allowing for features to be both low-level and high-level or abstract, potentially including safety-critical features, and ideally "covering" all the true underlying features learned by the model. Furthermore, explanations are sparse, increasing their human interpretability. However, the learned features are frequently not human-interpretable, leading to labeling errors and lower causal fidelity in practice. Furthermore, they require significant training and labeling data and compute, and there is no guarantee that a desired feature will exist in the SAE's dictionary, also preventing their utility in safety-critical domains. **Probing** allows for feature specification, given that a labeled dataset is available, but is prone to learning spurious correlations, leading to low causal fidelity and minimal steering utility. **Steering vectors** are very similar to probes in practice, but they require no optimization and are slightly less sensitive to correlations in the training data. As such, we believe that lens-based methods are most useful for providing high-fidelity explanations but are likely not effective solutions for steering in real applications, for which steering vectors and prompting are more promising but require careful oversight and refinement to ensure efficacy. SAEs offer a balance between interpretation- and control-based methods but also suffer from a union of their problems.

## 5 CONCLUSION

As language models are integrated into everyday aspects of life, whether benign or life-changing, being able to understand them and ensure they behave as expected is of utmost importance. While interpretability methods show great promise, the correctness of their interpretations is less clear. In this work, we emphasize that intervention is a fundamental goal of interpretability, both to evaluate the correctness of interpretability methods as well as their utility in real-world applications. By unifying and extending four popular interpretability methods — sparse autoencoders, logit lens, tuned lens, and probing — into an abstracted encoder-decoder framework, we enable structured interventions on human-interpretable features, mapping them back to model latent representations to directly control outputs. Through the introduction of two standardized metrics, the intervention success rate, and the coherence-intervention tradeoff, we provide a rigorous basis for evaluating the causal fidelity and utility of interpretability methods. Our findings reveal significant inconsistencies in intervention capability across different methods and models, with simpler lens-based approaches generally outperforming the more compute-intensive sparse autoencoders for simple, concrete interventions. Furthermore, non-interpretability-based approaches, such as prompting, perform best overall, demonstrating a significant gap in current interpretability approaches.

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

# A APPENDIX

## A.1 ADDITIONAL IMPLEMENTATION DETAILS: INTERVENTION HYPERPARAMETER $\alpha$

When intervening on $z$ to get $z'$ with Logit Lens, Tuned Lens, and SAEs, we set $z'_i = \alpha * max(z)$. For probing and steering vectors, $\hat{x}' = x + \alpha * v$, where $v$ is the steering vector or the weights of the linear probe. Note that $\alpha$ is a hyperparameter that must be tuned for each method and model, and thus cannot be used to compare the effects of interventions across methods. We record the values of $\alpha$ used in our experiments in Table 2.

Table 2: Values for hyperparameter $\alpha$ used to control intervention edit distance for each method and model.

| Method | GPT2-sm Layer 9 | Gemma2-2b Layer 20 | Llama2-7b Layer 18 |
|---|---|---|---|
| Logit Lens | [50, 70, 90, 110, 130] | [100, 130, 160, 200, 230] | [0.5, 3, 7, 11, 15, 19] |
| Tuned Lens | [20, 25, 30, 35, 40] | – | [1, 7, 11, 15, 19, 23] |
| SAEs | [3, 4, 5, 6] | [1, 2, 3, 4, 5] | – |
| Probing | [150, 200, 250, 300, 350] | [200, 250, 300, 350] | [10, 90, 110, 130, 150] |
| Steering Vectors | [2, 4, 6, 8, 10] | [3, 4, 5, 6] | [0.5, 3, 4, 5, 6] |

## A.2 ADDITIONAL IMPLEMENTATION DETAILS: SAE FEATURES

As described in Section 4.1, we use the sparse autoencoders hosted on SAELens and find the relevant features with Neuronpedia's exploration and search tools. We document all of the features we consider for each intervention topic in Table 3. Note that for some specified intervention topics, an exact feature match does not exist for the GPT or Gemma SAEs. As such, we either exclude that topic or consider the closest-related topic (such as "instruction related to yoga poses and their benefits" when what we would like is "references to yoga"). Many of these imperfect features still yield reasonable intervention success rates.

Table 3: Specific SAE features used for intervention on GPT2-sm and Gemma2-2b. The feature ids and their according Neuronpedia labels are provided.

| Intervention Feature | GPT2-sm Layer 9 Feature | GPT2-sm SAE Layer 9 Name | Gemma2-2b Layer 20 Feature | Gemma2-2b SAE Layer 20 Feature Label |
|---|---|---|---|---|
| San Francisco | 11233 | "mentions of the city of San Francisco" | 3124 | "references to San Francisco and related locations" |
| New York | 5831 | "references to the city of New York" | 3761 | "specific place names and geographical locations in New York" |
| beauty | 1805 | "words related to beauty or aesthetic appreciation" | 485 | "instances of the word "beauty" in various contexts" |
| football | – | – | 11252 | "references to football and baseball contexts" |
| pink | 2415 | "mentions of the word "Pink."" | 13703 | "references to the color pink and its various associations" |
| dogs | 12435 | "mentions of dogs or dog-related terms" | 12082 | "references to dog behavior and interactions" |
| yoga | – | – | 6310 | "instructions related to yoga poses and their benefits" |
| chess | 21685 | "mentions of the game of chess" | 13419 | "elements within the context of chess" |
| snow | 5053 | "references to snow-related terms" | 13267 | "references to snow and related terms" |
| coffee | 23472 | "references to coffee-related words" | 15907 | "references to coffee and related cafés or establishments" |

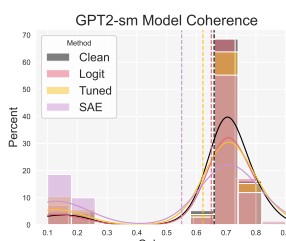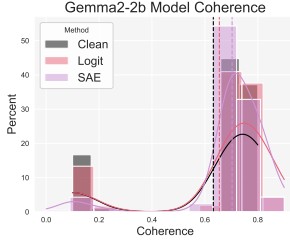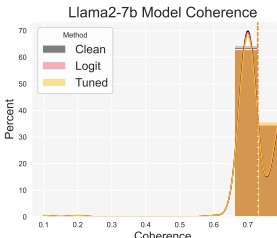

Figure 7: Histogram of coherence scores for clean model outputs (Clean) and for the models where $x$ is replaced by $\hat{x}$ without any intervention for Logit Lens, Tuned Lens, and SAEs. Dashed lines show the mean for each distribution.

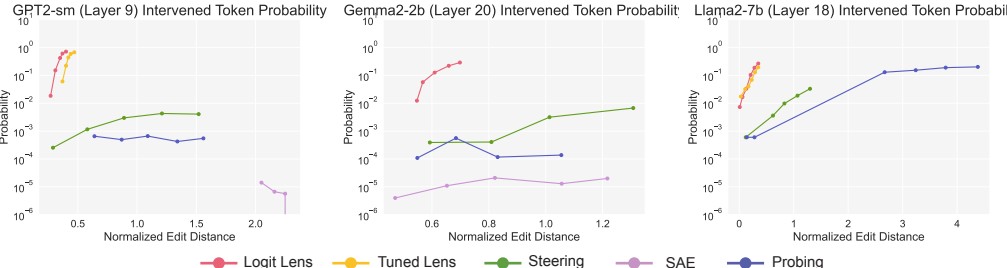

Figure 8: Evaluation of intervention success with respect to the probabilities of the tokens corresponding to the features intervened on for each method. Note that normalized edit distance is a proxy for intervention intensity that is comparable across methods.

### A.3 ADDITIONAL EVALUATIONS: COHERENCE OF METHOD OUTPUTS WITHOUT INTERVENTION

We measure the coherence of the outputs produced by replacing $x$ with $\hat{x}$, as shown in Appendix Figure 7, which we can compare to the baseline of the clean model outputs (labelled 'Clean' and shown in black). We find that the coherence of the outputs generated by the reconstructed latents generally matches the coherence of the clean model outputs. We use a deviation of $\pm 1$ around the mean of clean output coherence scores as a threshold for future evaluations, shown in the dashed lines.

### A.4 ADDITIONAL METRICS: INTERVENED TOKEN PROBABILITY

Please see Section 3.2 for more details. We measure the probability assigned to tokens relating to feature $i$ when intervening on feature $i$. As such, even if a model's output does not directly reflect interventions made to $z_i'$ due to sampling, we can measure if increasing the activation of feature $i$ results in any change to the model's output at all. We refer to this metric as **Intervened Token Probability**. Importantly, both of these metrics can be thought of as measuring the causality of the features highlighted by explanations.

Results for Intervened Token Probability are shown in 8, where we see that intervention with all methods across all models increases the probability of intervention-related tokens, even if the intervention does not succeed. We also note that there is a significant difference between the order of magnitude of the intervened token probability for sparse autoencoders, around $10e^{-5}$ and the rest of the methods, which range from $10e^{-4}$ to 0.5.

### A.5 ADDITIONAL METRICS: PERPLEXITY

As described in Section 3.2, we evaluate the perplexity of the intervened generated text to measure the utility of interpretability methods for targeted intervention in 9. We measure this perplexity with respect to a stronger language model than the one studied, in this case with Llama3.1-8b.

We find that the results for perplexity are generally unintuitive and do not align with the results for coherence. We hypothesize that perplexity is not a useful measure when text is extremely out-of-distribution with respect to normal text, and in particular when the text is highly repetitive. For example, if the same token is repeated 20 times, we (and other language models) might assume that the next 20 tokens would also be the same, resulting in a low perplexity even if the quality of the text is poor. As such, we do not consider these results to be particularly meaningful or significant.

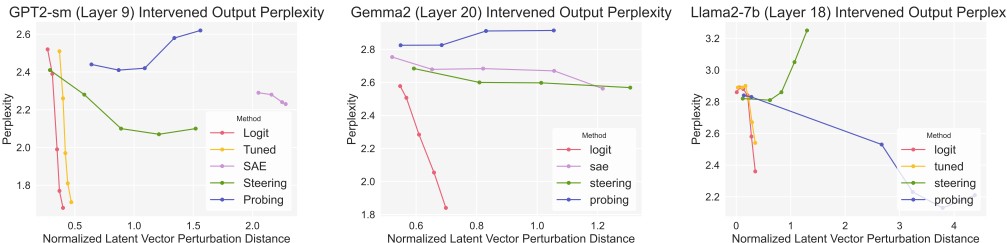

Figure 9: Analysis of perplexity of the intervened outputs, measured with Llama3.1-8b, as an alternative metric to Coherence. We find that perplexity does not align with Coherence, as highly repetitive sequences may have low perplexity despite being incoherent answers to prompts.

### A.6 INTERVENTION EFFICACY ACROSS MODEL DEPTH

In order to ensure the generalizability of the above results across layer depths, we repeat all experiments for each layer of GPT2-sm, as shown in Figure 10. However, due to some sparse autoencoder features only existing in some layers, we could only consider intervention topics { 'beauty', 'coffee', 'dogs'}. We hold the hyperparameter $\alpha$ that controls for intervention "strength" constant across all layers. Note that this is not equivalent to holding the normalized edit distance constant, as shown in the rightmost plot.

We find that layer depth seems to have minimal effect for SAEs and probing, with the exception of the first and last layers. For steering vectors, we observe a modest increase in intervention success rate with increased layer depth and a much more drastic increase in the success rate at later layers for Logit Lens and Tuned Lens. However, as we increase $\alpha$ significantly, we find that the curves for all three methods on intervention rate shift left until the pass rate is approximately 1 at all layers. Intuitively, this makes sense, as any edits to the residual stream at layer 0 will affect the residual stream at later layers. We note that these results highlight the need to tune the intervention strength for each method, each model, and each layer - limiting their ease of use.

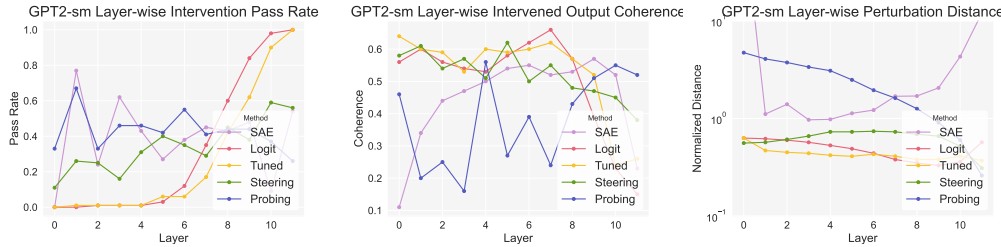

Figure 10: Analysis of intervention pass rate (left), coherence (middle) and edit distance (right) across all layers of GPT2-sm. We find that intervening at later layers is significantly more effective for Logit and Tuned Lens than earlier interventions, but probes, steering vectors, and SAEs are relatively invariant to the choice of layer.

### A.7 ADDITIONAL EXAMPLE OUTPUTS

We present additional examples of the output text for all intervention methods in Figures 11 and 12 for qualitative evaluation of intervention on the feature 'coffee' and 'San Francisco'. Examples for

the "Optimal intervention strength" (left column) were randomly chosen from the outputs where intervention succeeded and coherence was still relatively high. Examples for "Excessive Intervention" were randomly chosen from the outputs of the highest intervention strength tested (right column). Please see Section 4.4 for more.

| Method | Optimal Intervention Strength | Excessive Intervention |
|---|---|---|
| Logit | In my daydreams, I often find myself in a **coffee** shop, sipping a cup of **coffee** and reading a book. I love the smell of **coffee** and the taste of **coffee**.

(Edit Distance: 0.6, Coherence: 6, Model: Gemma2) | The joys and challenges of traveling alone include **coffee** shops, museums, **coffee** shops, **coffee** shops, **coffee** shops, **coffee** shops, **coffee** shops, **coffee** shops, **coffee** shops, **coffee**

(Edit Distance: 0.7, Coherence: 2, Model: Gemma2) |
| Probing | I have travelled far and wide, and, I have seen many different types of **coffee**. I have had the pleasure of trying some of the best **coffee** in the world. I have also had the

(Edit Distance: 0.8, Coherence: 5, Model: Gemma2) | Whenever I'm outdoors and in nature, I '4111111111111111111111111111

(Edit Distance: 1.1, Coherence: 1, Model: Gemma2) |
| SAE | I plan to write my Ph.D. dissertation on the topic of the relationship between the **coffee** and the **coffee** shop. I am interested in the history of the **coffee** shop, the history of the **coffee** shop

(Edit Distance: 0.9, Coherence: 5, Model: Gemma2) | In my previous life, I'm sure that I was a **coffee** shop barista. I' **coffee** is always fresh, the **coffee** is always fresh, and the **coffee** is always fresh. I's a great

(Edit Distance: 1.2, Coherence: 3, Model: Gemma2) |
| Steering | List some fun facts to share with kids.
 * \<strong\>The first **coffee** shop in the world was in 1977 in the city of Seattle, in the state of Washington.

(Edit Distance: 0.6, Coherence: 7, Model: Gemma2) | My favorite national park to visit is:  the \<strong\>\<em\>National Park:\</em\>\</strong\> \<strong\>\<em\>The\</em\>\</strong\> \<strong\>\<em\>\<strong\>\<em\>\<strong\>\<em\>\<strong\>\<em\>

(Edit Distance: 1.3, Coherence: 1, Model: Gemma2) |
| Tuned | Check out this haiku I wrote:
**coffee**, my dear friend
warming my hands and heart
in this cold world

(Edit Distance: 0.4, Coherence: 6, Model: Llama2) | The coolest wildlife I've ever spotted was a black **coffee coffee coffee coffee coffee coffee coffee coffee coffee coffee coffee coffee coffee coffee coffee coffee coffee coffee coffee coffee coffee coffee coffee coffee coffee coffee coffee coffee coffee coffee coffee**

(Edit Distance: 0.5, Coherence: 2, Model: Llama2) |

Figure 11: Example outputs with intervention on "coffee" feature.

| Method | Optimal Intervention Strength | Excessive Intervention |
|---|---|---|
| Logit | One dramatic impact of climate change on wildlife will be the spread of disease. The **San Francisco** Zoo is taking steps to prepare for the arrival of the first cases of disease in the San Francisco Bay Area.

(Edit Distance: 0.6, Coherence: 8, Model: Gemma2) | A book on art history that I found fascinating is Francisco Goya, Francisco Goya Francisco Goya Francisco Goya Francisco Goya Francisco Francisco Francisco Francisco Francisco Francisco Francisco Francisco Francisco Francisco Francisco

(Edit Distance: 0.7, Coherence: 2, Model: Gemma2) |
| Probing | Next month, I plan to travel to the United States to attend the 2016 International Conference on the History of the Book. The conference will be held in **San Francisco**, California

(Edit Distance: 0.7, Coherence: 8, Model: Gemma2) | A book on art history that I found fascinating is \<em\>The Modern Art Show: The 1932/33 Art/Alfar/Alfar/Alfar/Alfar/

(Edit Distance: 1.1, Coherence: 1, Model: Gemma2) |
| SAE | During my last work trip, I was able to visit the **San Francisco** Museum of Modern, and I was able to see the new exhibit, "The Last Resistance." The exhibit is a

(Edit Distance: 0.9, Coherence: 8, Model: Gemma2) | Some examples of eco-friendly destinations and tips for minimizing your travel footprint are \<strong\>San Marin\</strong\> \<strong\>Island\</strong\>, \<strong\>Tib\</strong\> \<strong\>Travel\</strong\>, \<strong\>The\</strong\> \<strong\>Golden Emb Emb Emb\</strong\>

(Edit Distance: 1.2, Coherence: 1, Model: Gemma2) |
| Steering | My favorite childhood memory is of my parents taking me to the **San Francisco** Exploratorium. I remember being amazed by the Exploratorium's Exploratory Playroom, which was a

(Edit Distance: 0.6, Coherence: ,8 Model: Gemma2) | In my daydreams, I like to imagine that I'm a 19th-sf Francisco Francisco Francisco Francisco Francisco Francisco Francisco Francisco Francisco Francisco Francisco Francisco Francisco Francisco Francisco

(Edit Distance: 1.3, Coherence: 2, Model: Gemma2) |
| Tuned | Check out this haiku I wrote:
**San Francisco**'s hills
Steep and winding, a challenge
To walkers, bikers too

(Edit Distance: 0.4, Coherence: 7, Model: Gemma2) | My favorite song from the 21st century is "Ho Hey" by **San Francisco**-based indie rock band The **San Francisco** Francisco Francisco Francisco Francisco Francisco Francisco Francisco Francisco

(Edit Distance: 0.5, Coherence: 3, Model: Llama2) |

Figure 12: Example outputs with intervention on "San Francisco" feature.

## B  Rebuttal Experiments

In this section, we provide additional experiments as requested by reviewers during the rebuttal period. First, we present results on the larger and more recent model Llama3-8b (Instruction Tuned) in B.1. We also evaluate an alternative metric to Coherence that measures the number of grammatical errors in the intervened output via a rule-based grammar checker in B.2. Finally, we present experiments for more complex features, including responding in the French language and responding by yelling in B.3

### B.1  Intervention Experiments on Llama3-8b

We present additional results on Llama3-8b (Instruction Tuned), the highest-performing and most recent model for which we were able to find a publicly available, trained, and labeled SAE on SAELens and Neuronpedia Bloom (2024b); Lin & Bloom (2023). In particular, we repeat the experiments from 4 using the same prompts and intervention features, and present a similar figure to Figure 4 evaluating the coherence of the intervened output text with respect to the intervention success rate for Logit Lens, Tuned Lens, Steering, SAEs, Probes, and the Clean LLM Baseline in Figure 13. We find that the results for Llama3-8b largely match those of Llama2-7b, with Logit and Tuned Lens outperforming the other three interpretability and intervention methods. Furthermore, all methods still present strong tradeoffs between intervention success and output coherence and underperform the simple prompting baseline.

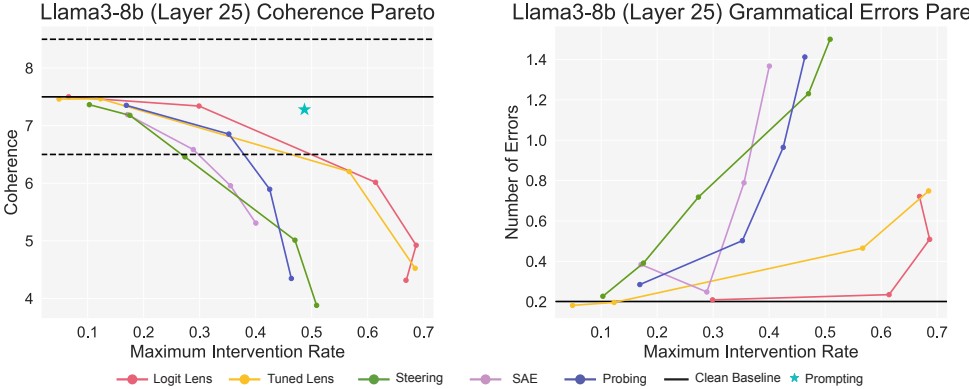

Figure 13: Intervened output coherence (left) and number of grammatical errors (right) measured with respect to intervention success rate for intervention on Layer 25 of Instruction Tuned Llama3-8b. We find that results are consistent with those for Llama2-7b.

### B.2  Coherence via Rules-Based Grammar Checker

We evaluate an alternative metric to Coherence that measures the number of grammatical errors in the intervened output via a rule-based grammar checker. In particular, we use LanguageTool to determine the number of errors in each output, which has over 6,000 rules relating to grammar, typos, capitalization errors, and more. Results are presented for Llama3-8b (Layer 25) for direct comparison with those for Coherence in Figure 13. Overall, we find highly consistent results between Coherence and the number of grammatical errors, where Logit Lens and Tuned Lens have the fewest number of errors in their intervened outputs at high intervention rates.

### B.3  More Complex Features: French and Yelling

We additionally consider two more intervention features: responding in French and responding by yelling in upper-case text. We use the same prompt dataset as in the experiments of the main paper (detailed in Section 3.2), which consists of only English prompts with standard capitalization schemes (proper nouns and the first letter of each sentence). We consider these additional features due to their availability in the Gemma2-2b SAE used in Section 4. While more safety-critical or

high-stakes features would be interesting to consider, we were unable to find such features that successfully steered the model during the rebuttal period. We further note that both French and yelling are complex features because they cannot be captured by a small set of words or tokens.

**Implementation details.** Datasets used to train steering vectors and probes were generated in a similar way to those for the low-level features. Specifically, we generate 200 simple sentences in English with the help of an LLM and manually inspect them to ensure correctness. We then translate those sentences into French or simply capitalize the sentences to get contrastive datasets for the steering vectors and probes. Both probes achieved perfect test accuracy. We intervened on the "French text" and "uppercase text" features of the same Layer 20 Gemma2-2b residual stream SAE with 16k features used in Section 4. Note that these two features are advertised as preset features on the Steering page of Gemma Scope/Neuronpedia as "French mode" and "YELLING mode" Lin & Bloom (2023).

Given that these features are not naturally present in the dictionary of Logit Lens, we "learned" the set of Logit Lens features to intervene on by picking the 200 most common upper-case tokens from the steering vector/probe datasets for the 'yelling' concept and by picking the 200 tokens with the highest cosine similarity to the learned 'French' steering vector. Note that while the token features for 'yelling' were still highly interpretable, the features similar to the 'French' steering vector were not, and often did not intuitively relate to the French language. Thus, we would not recommend using Logit Lens to intervene on the 'French' concept in practice, but simply test this method for completeness. Tuned Lens is not considered because there is no publicly available trained Tuned Lens for Gemma2 as of yer.

Results for intervention on these two features are shown in Figure 14, where we see that Steering vectors and Probes generally perform the best. As expected, Logit Lens does not perform well due to the intervention features not being present in Logit Lens's dictionary. We further find that SAEs perform very poorly, despite these features being chosen directly from the SAE dictionary. Overall, the results on these two complex features align with the results and conclusion drawn in Section 4.6, where we note that lens-based methods are most useful for providing high-fidelity explanations but are not effective solutions for steering in real applications with complex features, for which steering vectors and prompting are more promising but require careful oversight and refinement to ensure efficacy.

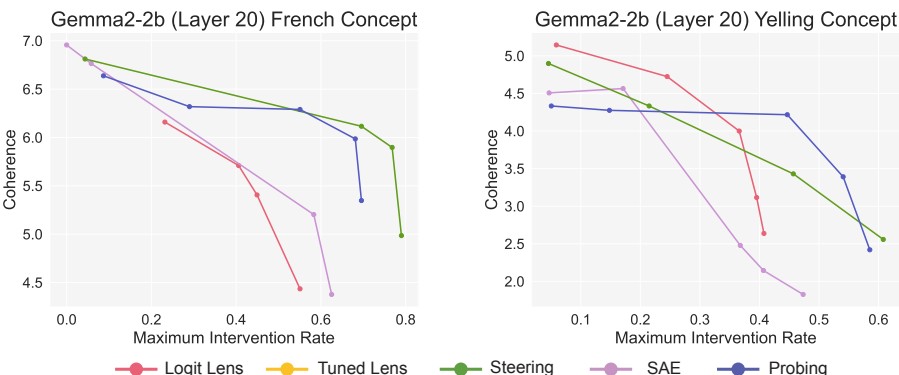

Figure 14: Intervened output coherence measured with respect to intervention success rate for intervention on Layer 20 of Gemma2-2b for two complex concepts: French (left) and Yelling (right). We find that supervised methods (Steering vectors and probes) are the highest performing, but still present drastic tradeoffs between intervention and coherence.

