# OpenReview forum: "Towards Unifying Interpretability and Control: Evaluation via Intervention"
_ICLR.cc/2025/Conference — Submitted to ICLR 2025_

### Official Review · Reviewer_oPk4 · 2024-10-19

**Soundness:** 3
**Presentation:** 3
**Contribution:** 3
**Rating:** 6
**Confidence:** 2

**Summary:**

To bridge the gap between research and techniques focused primarily on one of either understanding or control in LLMs, this paper presents a framework to unify interpretability and control through intervention.  They propose an encoder-decoder framework to enable both explanations from and interventions on latent representations, mapping human-interpretable features back to the model's latent space.  Two metrics, intervention success rate and coherence-intervention tradeoff, are introduced to assess the methods in a more holistic and direct way than was previously possible. The results show that while current methods can enable intervention, they are inconsistent across models and features, and can compromise model coherence.

**Strengths:**

The paper demonstrates originality by unifying interpretability and control, addressing a key gap in popular research directions where most work focuses on either understanding or steering models. Its introduction of a unified encoder-decoder framework and new evaluation metrics (intervention success rate and coherence-intervention tradeoff) provides a novel perspective for benchmarking existing interpretability techniques more uniformly and on a more level playing field. The experiments evaluate multiple popular methods across various models, exposing the relative strengths and weaknesses of different approaches.  The method and evaluation framework are clearly explained, and supported by a useful Figure 1.  This paper seems like a significant contribution to the field, offering a framework to unify the pursuit of interpretability and control, an obvious yet elusive goal in LLM interpretability.  I believe it can stand as a valuable resource for ongoing research on model safety and alignment.

**Weaknesses:**

The primary weakness of the paper is the lack of extension to complex, real-world applications.  Evaluation is limited to relatively simple interventions, such as word-level steering (e.g., 'coffee' or 'yoga') w.r.t. concrete concepts, which may not fully capture the challenges of more abstract, high-stakes tasks such as ethical decision-making, bias reduction, or truthfulness.  While this limitation is addressed acknowledged by the authors, and common in this research area, it does stand as a challenge in being able to draw any strong conclusions from the experiments.

**Questions:**

It seems that the experiments rely heavily on data/prompts generated by LLMs.  Is this data inspected by humans, or otherwise validated for correctness?

---

> ### Author Response · Authors · 2024-11-20
>
> Thank you for your suggestions! We appreciate your feedback on how to improve this work and address your concerns below.
>
> **Extension to complex, real-world applications and abstract tasks.** While we agree that empirical results for more abstract concepts and applications would be highly interesting, we are limited by the feature spaces of the interpretability methods evaluated. In particular, we cannot specify the features learned by SAEs or encoded by Logit Lens or Tuned Lens. Thus, we can only evaluate them for the features they provide us, which are generally low-level token- or word-related features. Furthermore, we believe that if an explanation method doesn’t admit intervention on lower-level features, we cannot expect it to work on the high-level features we might care about in downstream tasks, thus making high performance on low-level features imperative. Additionally, we note that it is easier to measure low-level features due to their objective evaluation (e.g. labeling an output as referring to “yoga” is much easier than labeling an output as “safe”).
>
> We include additional results for two more complex concepts: (1) responding in the French language despite being prompted in English and (2) responding by “yelling” (using all upper-case text) in Section B.3 of the Appendix (page 19). Both of these concepts are not tied to specific tokens, thus making them relatively complex. We find that supervised methods (probes and steering vectors) perform best on these complex features, aligning with our expectations and results in Section 4.6. Even so, these methods are highly suboptimal, and intervention often causes drastic loss in model coherence, despite the studied features being easy to encode through prompting. While results for safety-critical features would be interesting, we were unable to find such features that permitted steering in labeled and publicly available SAEs, and we further believe this application to be beyond the scope of this work.
>
> **Human inspection of data/prompts.** All of the training datasets and prompts were evaluated by humans and are provided in our accompanying repository for inspection by the reviewers or the public.

---

> > ### Comment · Reviewer_oPk4 · 2024-11-21
> > **Reviewer Acknowledgement**
> >
> > I think the authors for their responses.  I will maintain my score.

---

> > > ### Author Response · Authors · 2024-11-24
> > >
> > > Thank you for your engagement and response! Please let us know if you have any remaining unanswered questions. Otherwise, if we have sufficiently answered all of your questions and concerns, we kindly request you to consider increasing your score.

---

### Official Review · Reviewer_1BCv · 2024-10-26

**Soundness:** 2
**Presentation:** 2
**Contribution:** 3
**Rating:** 6
**Confidence:** 3

**Summary:**

The proposed approach adopts an "intervention" based perspective to LLM interpretability. Two metrics for evaluating mechanistic interpretability approaches are proposed: a "intervention success rate" that quantifies how often changes in an interpretable feature yield desired changes in model outputs, and a "coherence-intervention tradeoff" that quantifies tradeoffs between intervention success and the coherence of model outputs. Extensive experiments demonstrate that different interpretability techniques yield interventions with varying  differing success rates and levels of output-coherence after the intervention.

**Strengths:**

* The problem area is well-motivated and timely — indeed, there's a gap in evaluating the "quality" of different interpretability techniques for LLMs. The proposed approach fills the gap by proposing concrete quality metrics to that end, and the experiments comprise multiple proofs-of-concept for how one could use the proposed metrics to evaluate interpretability techniques.
* Some of the proposed metrics are intuitive and easy to measure. I particularly like the "intervention success" metric — while simple, it's a really intuitive proxy for whether an interpretability technique actually provides an "intervene-able" interpretation.
* Contributing a benchmark of open-ended prompts from a variety of domains is also valuable — it's not simply another "task/capability" for LLMs to solve, but a natural test-bed for diversifying the space of possible interventions under the proposed framework.

**Weaknesses:**

Major
* [W1] Comparability of approaches: Given that the learned $D$ has different columns depending on the approach (esp. SAE vs. the others), it is not immediately clear how one can compare different approaches directly. Furthermore, it's unclear if low "intervenability" (success rate and coherence) as measured by the proposed approach is attributable to a poor process for learning $D$, or the inability of the base LLM to "steer" along the relevant feature. More generally, the metrics could reflect either the behavior of the base LLM or the feature extraction approach.
* [W2] Evaluation metrics: I like the definition of intervention success, but coherence is much less clear. The explanation in the paper is "This can be done by any oracle, such as a human or another capable LLM" — but these approaches have disadvantages, since human annotation may be expensive, and an LLM-as-judge approach is not necessarily well-calibrated w.r.t. human annotations.
* [W3] Undefined notation: $x$ is not defined at the top of 3.1, and $\theta$ is not defined (L200) in the explanation of probing. Again, I can guess what these mean from related works/Figure 1, but there's a high risk of misunderstanding.

Minor
* Lack of clarity in Figure 1: I know that symbols such as $D$ are defined later in the manuscript, but given that we encounter figure 1 at the top of page 2, it's a difficult to understand what the columns of the table are. This also makes it hard for me to appreciate the extent of the papers' contributions. I think the Table in Fig. 1 would be valuable later in the paper, after some expectations have been set about  desiderata for LLM interpretability evaluations.  At the very least, I think I need to see Section 3.1 before seeing the Table.
* Limited to open-weight models: while this limitation is shared by many approaches for interpretability/controllability, it would be nice to hear about potential extensions to models where the only access to a model is via API. For example, one could still intervene at the prompt level. Ultimately, this is very minor and wouldn't sway my score.

**Questions:**

Answers to the following questions would be most likely to sway my score:
* **Re [W1]:** Could the authors provide more details on how one can make SAEs comparable to the other approaches for the purposes of comparing interprtability techniques (i.e., expand on the description in 4.1)?
* **Re [W1]:** In the experiments, high intervenability is equivalent to "frequency of word in sentence" due to the setup. Doesn't this give an unfair advantage to LogitLens/approaches where the feature space under interpretation is already the vocabulary?
* **Re [W2]:** How might one design a more systematic/rigorous metric for coherence (*e.g.*, why not simply use rules-based tools to check for grammatical errors)? Are "grammar" and "comprehension" the only relevant dimensions for coherence (as used in the LLM-as-judge prompt in the paper)? How did/how might the authors plan to validate their coherence metric?

---

> ### Author Response · Authors · 2024-11-20
>
> Thank you for your suggestions! We appreciate your feedback on how to improve this work and address your concerns below.
>
> **Comparability across approaches.** Despite the different dimensionalities or columns of the dictionaries learned by each interpretability method, all methods attempt to explain the same latent vectors by using similar features in the same space. More specifically, a single feature of a trained SAE might predict the presence of “San Francisco” in the model’s input text, as would a probe trained on a dataset that encodes for “San Francisco”, and high logit values assigned to San Francisco’s corresponding tokens in Logit Lens and Tuned Lens. Thus methods are still comparable despite their different feature spaces because they all operate on the same latent space. Furthermore, we note that the discrepancies between Ds is what warrants a unified framework for evaluation in the first place - if different methods are all attempting to explain the same feature using different Ds, how can we assess which one is most correct or faithful or which one is best for intervention and control?
>
> **Steerability / behavior of the base LLM.** In order to ensure that the base LLM itself is steerable, we consider the baseline of simply prompting the LLM to increase the feature of interest (e.g. we ask it to talk more about coffee), as denoted by the teal stars in Fig. 4. If the model is able to be prompted towards a certain behavior, we believe that steering and intervention should also be effective. We found that prompting was more effective than all interpretability methods at intervening on model behavior, thus indicating that poor intervenability is more likely due to the interpretability methods than the LLM.
> Additionally, we craft our prompt dataset such that steering should be feasible given the open-ended nature of the prompts and the simple intervention topics considered.
>
> More generally, we note that if an explanation method claims that the model is encoding information corresponding to some feature, then steering the model to further increase the presence of that feature should be possible. If such steering were not possible and the model does not actually encode for the feature, the interpretability method itself is likely incorrect and should not have provided an explanation including that feature in the first place.
>
> **Definition of coherence.** We agree that using humans and LLMs to measure coherence both present different limitations, and we will be sure to expand on the discussion of this limitation in the final paper. However, we note that even if there is slight misalignment between the absolute values of coherence as rated by humans versus coherence as rated by an LLM, what matters more is the relative coherence between the clean LLM’s text and the intervened outputs. We have provided baselines of the coherence of clean outputs in the black lines in Figures 3 and 4 to better calibrate to the LLM rater’s scale of coherence. Furthermore, to validate using LLMs to measure coherence, we present results using a rules-based grammar checker in the supplementary materials of the rebuttal (also discussed below in point ‘Rigorous coherence metric.’ We note that prior literature frequently uses LLMs-as-judges due to the expense of human annotators, which is what motivated our use of LLMs to measure coherence.
>
> **Undefined notation.** Thank you for noting this. We will be sure to clarify this notation in the final version of the paper, where x is the latent representation of the model and theta corresponds to the learned weights of the probe.
>
> **Fig. 1 Clarity.** Thank you for bringing this to our attention. We are happy to make the changes suggested by the reviewer and will accordingly move the table further down in the paper.

---

> > ### Author Response · Authors · 2024-11-20
> >
> > **Open-weight vs API access models.** We agree that extensions to API access models would be interesting; however, the interpretability methods that we evaluate here all rely upon the hidden representations of the model, which are not provided by API access. As such, we are limited to exploring open-weight models. We do, however, provide steering via prompting baselines for the instruction-tuned models (Llama and Gemma) in the paper (as denoted by the teal stars in Fig. 4), where we see that prompting is more effective than all interpretability methods evaluated.
> >
> > **How to make SAEs comparable / 4.1.** We make SAEs comparable to the other methods in this work by intervening on the sparse feature activations produced by the SAE. Specifically, we first identify the exact features we wish to intervene on by searching through the feature labels for the given SAE on Neuronpedia, which provides autointerpreted labels for each feature of the SAEs we evaluate in the paper. On Neuronpedia, we can look up the id or index of each feature we consider in the encoder and decoder matrices of the SAE. We then encode the LLM’s latent representation with the SAE, identify the feature activation corresponding to the feature of interest by using the id from Neuronpedia, and then modify that activation to produce our edited explanation. We can then decode this edited explanation back into the latent space of the language model by using the SAE decoder. This edited model latent can then be passed through the remaining layers of the language model to get the intervened output.
> >
> > **Advantage to Logit Lens.** We note that many features learned by SAEs are low-level features corresponding to tokens, words, or short phrases, and are thus similarly in the “vocabulary space,” such as a feature labeled as “mentions of the city of San Francisco” (Appendix Table 3.). These low-level features are also easily captured by steering vectors and probes, and thus should not provide Logit/Tuned Lens with any advantage. Furthermore, many of the features we tested are actually multiple tokens long (e.g. San Francisco), and thus are not directly encoded by lens-based methods. As such, we believe the low-level features tested allow for a fair comparison across methods and are particularly valuable (as noted by reviewer QT3J) given that if an explanation method doesn’t admit intervention on lower-level features, we cannot expect it to work on the high-level features we might care about in downstream tasks. Thus, we believe that high performance on low-level features is imperative and a prerequisite to intervening on more abstract or high-risk features.
> >
> > However, we definitely agree that the converse is true - lens-based methods are at an extreme disadvantage when considering more abstract or high-level features that cannot be captured by a few tokens. We explore the concepts of (a) responding in the French language and (b) responding by yelling in upper-case text in the provided supplementary materials for the rebuttal (Appendix Section B.3, pages 18-19), where we see that steering vectors and probes generally perform the best but are still highly suboptimal.
> >
> > **Rigorous coherence metric.** We thank the reviewer for this suggestion. We present experiments using a simple rule-based checker from the NLP library LanguageTool to check for the number of grammatical errors in outputs in Section B.2 of the Appendix (page 18). We find that the results and conclusions are consistent between both metrics of coherence and number of errors, where Logit- and Tuned-lens are generally optimal. We also find that coherence and number of errors have a strong negative correlation across methods and models, as expected. We will be sure to include these results in the final paper.

---

> > > ### Comment · Reviewer_1BCv · 2024-11-22
> > >
> > > Thank you for the thoughtful responses! I think some of my specific concerns re: evaluation are addressed, but, having read the other reviews & the rebuttal comments, I think there are some claims/discussion of causal methods that don't seem quite right.
> > >
> > > First, I'm in partial agreement with Reviewer LqMe's assessment. I had missed the issue with the counterfactual prompts, and the prompt examples provided in the rebuttal response still change more than the target attribute; i.e., in addition to changing "dog" to a different animal, other features (e.g., the "action" taken by the other animal) are also swapped. So it is a little bit unclear whether the steering "directions" that arise from such data actually reflect the concept of interest, rather than some potential confounding. Second, I think there a false dichotomy between the proposed approach and causal mediation analysis — I don't find the goals to be opposite. The proposed approach, in attempting to compare approaches that extract some "basis" of human-interpretable features, seem to be a special case of 4.3 of [1].
> > >
> > > Ultimately, for these reasons, I've decided to keep my original score.
> > >
> > > [1] Mueller, Aaron, et al. "The quest for the right mediator: A history, survey, and theoretical grounding of causal interpretability." arXiv preprint arXiv:2408.01416 (2024).

---

> ### Author Response · Authors · 2024-11-24
>
> **Counterfactual prompts.** We note that the prompts used to train the steering vectors and probes are not meant to be *counterfactual* but *contrastive*, as is standard for the steering vector method we implement (Contrastive Activation Addition [1,2]). We use the term contrastive in the paper and make no claims of counterfactuality because it is difficult, if not impossible, to create the types of counterfactual datasets the reviewer is suggesting. For example, if we take 10 sentences about San Francisco, it is likely that replacing the name “San Francisco” with any other city name would yield a factually incorrect sentence (e.g. “San Francisco is home to the Golden Gate Bridge” → “New York is home to the Golden Gate Bridge”). As such, we cannot only edit the phrase of interest to create a counterfactual prompt, but we must instead generate a contrastive prompt that does not contain the same feature. We apologize for not correcting reviewer LqMe’s use of the word counterfactual. We further note that this issue of potential confounders or correlations in the data trained to use steering vectors and probes is why they are fundamentally less causal in nature, as is frequently noted in past literature, including many of the citations in [3]. We acknowledge this in our paper, including in Section 4.6.
>
> **Causal mediation analysis**. To clarify any confusion regarding this topic, in our discussion with reviewer LqMe, we “acknowledge that Intervention Success Rate can be thought of as an extension of IIA to feature-based interpretability methods.” We think of our work as working towards the suggestions for future work noted in Section 7.2.4 of [3], particularly an evaluation of interpretability methods for intervention. As noted by [3], many of the methods in 4.3 of [3] that extract bases of human-interpretable features are not systematically evaluated for intervention, and in fact these are the very methods that we evaluate in our paper (cited in [3] as Turner et al. (2023), Chen et al. (2024b), Templeton et al. (2024), Cunningham et al (2024)). As we noted to reviewer LqMe, we will be sure to include further discussion of causal mediation analysis in the final work, which was previously limited due to space constraints. We reiterate that the main goal of our work is to evaluate existing methods empirically, and that our framework is simply a method to standardize intervention across methods with different encoders, feature spaces, and decoders. We do not claim to be the first to propose evaluation via causal intervention, but that to our knowledge, we are first to present a standardized, systematic empirical study testing various existing methods for this ability.
>
> Furthermore, we note that [3] was only put on ArXiv in August 2024, and we were unaware of this work at the time of submission. As according to ICLR policy, works published after July 1, 2024 and any non peer-reviewed conference proceedings or journals need not be cited by authors, as they are considered “contemporaneous.” Nonetheless, we will be sure to discuss this concurrent work in the final version of our paper.
>
> [1] Panickssery, Nina, et al. "Steering llama 2 via contrastive activation addition." arXiv preprint arXiv:2312.06681 (2023).
>
> [2] Ghandeharioun, Asma, et al. "Who's asking? User personas and the mechanics of latent misalignment." arXiv preprint arXiv:2406.12094 (2024).
>
> [3] Mueller, Aaron, et al. "The quest for the right mediator: A history, survey, and theoretical grounding of causal interpretability." arXiv preprint arXiv:2408.01416 (2024).
>
> Please let us know if you have any remaining questions. Otherwise, if you feel your concerns were adequately addressed, we ask you to consider raising your score. Thank you!

---

> > ### Comment · Reviewer_1BCv · 2024-11-24
> > **Will update score to marginal accept (i.e., has significant limitations but good idea worth sharing w/ community)**
> >
> > Ah, thanks, that clears up some of the confusion around the "counterfactual"/causal-adjacent language. I've tried to think some more about the core contributions of the work, and whether the limitations are dealbreakers. As such, I've carefully re-read the paper and the associated reviews.
> >
> > The primary value of this paper in its attempt to stake out the following claim: "For a mechanistic interpretability technique to be useful, interventions in its feature basis *must necessarily* result in some change in the LLM output grounded in the feature of interest."
> >
> > The main limitations of the analysis in the paper is that it sticks to simple/word-level goals, that some definitions lack clarity (i.e., coherence; mostly addressed in rebuttal), and that the latest/largest models are not used. The first limitation is fairly large in my opinion, but I know that the paper's line of analysis on simpler features is likely a prerequisite to analysis of more complex features. The last limitation is moderate: the main "star of the paper" is the underlying mech interp technique rather than the underlying model. In my opinion, primary question of interest in analyzing larger models for this paper would be whether existing mech interp techniques are "biased" towards some size of models; i.e., do the weights of larger/smaller models yield more "intervenable" features. This could've been tested on different variations of the same model family (e.g., different sizes of GPT2), but I didn't think of this experiment in my initial review (and have no expectation of results here given the short timeframe). Weaknesses raised by other reviewers seem to be addressable at camera-ready time (e.g., adding citations/fixing typoes), or points of clarification.
> >
> > I'll also acknowledge that an important strength is that the proposed technique can be applied *out-of-the-box* to any mech. interp. technique that spits out a basis of features (i.e., most of them). The analysis of different interpretability techniques, highlighting some limitations of each, is of potentially independent interest to the mech interp community.
> >
> > So I see this paper as presenting a "usefulness prerequisite" for mech. interp techniques, rather than an end-all-be-all for gauging whether a mechanistic interpretability technique is "good" (in any sense). I was initially concerned that the base LLM's steerability could confound this measurement, but the argument about prompt-based interventions as a substitute (which doesn't exactly have an interpretable component) resolves this concern — it establishes the *existence* of some implicit "direction" where that feature lives.
> >
> > Thus, I don't actually find the lack of focus on complex/real-world tasks to be cause for outright rejection — i.e., the main contribution isn't really about the model or the underlying task. However, it's important to acknowledge that further analysis of such would significantly strengthen the paper.
> >
> > Based on this, I think I can justify raising my score to a marginal accept (i.e., good idea, with significant limitations that aren't paper-breaking). I think improved clarity in writing would have helped me come to this conclusion about the paper much quicker, but it was OK as-is.

---

### Official Review · Reviewer_LqMe · 2024-11-02

**Soundness:** 1
**Presentation:** 2
**Contribution:** 1
**Rating:** 3
**Confidence:** 4

**Summary:**

This paper aims to unify a few interpretability methods (SAEs, logit/tuned lens, probing) via a metaphor to encoder-decoder architectures. Specifically, the paper articulates what the "encoder" and "decoder" are for each of these interpretability methods. Explanations are evaluated by intervention-based evals: "intervention success rate" and "intervened token probability". Lens-based methods are claimed to achieve "more simple, concrete" interventions.

**Strengths:**

The idea of inverting the explanations in order to get a "natural" notion of intervention seems promising, especially if it is supplemented with theoretical analysis for why this is better than existing intervention-based evals.

**Weaknesses:**

While the proposed framework is interesting, this paper is not quite ready for acceptance at ICLR 2025; it overall appears incomplete and rushed.

1. Missing large areas of related work, which significantly weakens the claim of "unification", and undermines some claims of novelty.

- There is a whole subfield of intervention-based evals for interpretability; see the review paper below. This undermines the novelty of "we propose intervention as a fundamental goal of interpretability and introduce success criteria to evaluate how well methods are able to control model behavior though interventions" (Abstract, lines 18-20).

Mueller, A., Brinkmann, J., Li, M., Marks, S., Pal, K., Prakash, N., Rager, C., Sankaranarayanan, A., Sen Sharma, A., Sun, J., Todd, E., Bau, D., & Belinkov, Y. (2024). The quest for the right mediator: A history, survey, and theoretical grounding of causal interpretability. arXiv. https://arxiv.org/abs/2408.01416

- The closest citation to this field is Geiger et. al. (2021) in line 127, which is neither recent nor representative of causal abstractions.

- Consequently, the "intervention success rate" of Section 3.2 (which is barely defined, in lines 112, 242-244) seems quite similar to Interchange Intervention Accuracy, which has already been well-studed: see for instance

Geiger, A., Lu, H., Icard, T., & Potts, C. (2021). Causal abstractions of neural networks. arXiv. https://arxiv.org/abs/2106.02997

Geiger, A., Ibeling, D., Zur, A., Chaudhary, M., Chauhan, S., Huang, J., Arora, A., Wu, Z., Goodman, N., Potts, C., & Icard, T. (2024). Causal abstraction: A theoretical foundation for mechanistic interpretability. arXiv. https://arxiv.org/abs/2301.04709

Geiger, A., Wu, Z., Potts, C., Icard, T., & Goodman, N. (2024). Finding alignments between interpretable causal variables and distributed neural representations. arXiv. https://arxiv.org/abs/2303.02536

2. The evaluation metrics (Section 3.2, lines 237-249) lack crisp definitions, despite being listed as a main contribution.

3. Experiments and figures appear rushed.

- The captions of Figures 3 and 4 lack detail.

- It's not clear how many methods are tested, and different parts of the paper state different things. There are only 3 methods in Table 1; line 291 in the Experiments section says there are 5 methods; line 531 in the Conclusion says 4 interpretability methods.

- The prompts in lines 316-318 are not counterfactual: they change more than the target attribute.

- The data (lines 300-308) is quite restrictive, which may explain the 100% training and test accuracy of the probes (line 320).

- What exactly is the function used to measure coherence? Coherence does not appear to be defined anywhere: in line 388 it is simply stated that "we measure coherence ...as a function of the normalized latent edit distance...". What function exactly?

4. The encoder-decoder framework isn't motivated; assumptions are not discussed

- It's not clear (e.g. lines 82-88) what the encoder-decoder framework buys over e.g. mediation analysis.

In sum, this appears to be a preliminary research draft more appropriate for a conference workshop.

**Questions:**

1) Can you please position this paper relative to causal interpretability methods discussed in the review (Mueller 2024) cited above?

2) Is the goal of this paper to unify "interpretability" and "control" as mentioned in the title, or "unifying and extending four popular interpretability methods" (line 530-531)? These are very different!

---

> ### Author Response · Authors · 2024-11-20
>
> Thank you for your suggestions! We appreciate your feedback on how to improve this work and address your concerns below.
>
> **Related work, novelty, and position relative to causal interpretability methods.** We thank you for pointing us to this review paper! Mueller et al. (2024) succinctly describes many of the motivations for our paper - mainly that while mechanistic interpretability began as a field that leveraged causal interventions to create mechanistic or causal abstractions of models, many popular interpretability methods now aim to create human-interpretable feature-based explanations of model representations by relying on correlational statistics rather than causal theory (e.g. Sparse Autoencoders, Logit Lens, probes). This is frequently noted in Mueller et al. (2024) and Saphra et al. (2024), and both papers call for better evaluation and more standard benchmarks for these methods (see Section 7.2 of Meuller et al. where they note that "currently, most studies develop ad-hoc evaluations” and that more "standard benchmarks for measuring progress in mechanistic interpretability" are needed). We believe this work to be a step towards such systematic evaluations.
>
> We apologize for any confusion our wording may have caused, as we did not intend to say that we are the first to consider interventions for interpretability, but that to the best of our knowledge, we are first to present a systematic evaluation of various popular interpretability methods for intervention. We will be sure to update our language and make more references to past mechanistic interpretability work that focuses on causal interventions, including the citations provided by the reviewer, which were unfortunately previously omitted due to limited space.
>
> **Interchange intervention accuracy.** Interchange Intervention Accuracy measures the extent to which the functionality of one model can be induced in another model, when their latent variables or activations are interchanged. This allows for making a causal claim regarding the effect of the latent variables. In contrast, Intervention Success Rate focuses on interventions made to human-interpretable explanations of model latent representations. While both metrics aim to measure causal effects on the outputs of the model, the input space for both varies quite drastically. Furthermore, while interchange intervention works on uninterpretable latent vectors, our intervention success rate is applicable to only “interpretable” latent vectors, allowing us to evaluate their interpretability and utility for model intervention. We are happy to note the similarities between Intervention Success Rate and IIA in the final paper, and we acknowledge that Intervention Success Rate can be thought of as an extension of IIA to feature-based interpretability methods.
>
> **Definitions of evaluation metrics.** We apologize for the brevity of the descriptions of each evaluation metric due to the space constraints of the paper. We will provide more rigorous definitions of the metrics in the final version of the paper, including the additional coherence metric of the number of grammatical errors (proposed by reviewer 1BCv).
>
> **Number of methods.** In total, we evaluate four interpretability methods: {SAEs, Logit Lens, Tuned Lens, probes} and one control (but not interpretability) baseline: {steering vectors}, There are only three methods in Table 1, as noted in lines 349-351, because only SAEs, Logit Lens, and Tuned Lens can be used to fully reconstruct the model’s latent vectors. As such, we cannot perform the reconstruction sanity check on steering vectors and probes, and they are omitted from the table.
>
> **Counterfactual prompts.** We apologize for the confusion. The prompts provided in 316-318 are not the counterfactual prompts themselves but rather the prompts we used to generate the counterfactual datasets with an LLM. The full counterfactual datasets for each feature used to train the steering vectors and probes are provided in the code repository given in the supplementary material. For example, some positive prompts for the “dog” feature are {“The dog jumped into the pool,” “The dog howled at the moon,” “The dog had expressive eyes”} and some negative prompts are {“The pelican dived into the water,” “The wolf howled at the moon,” “The owl blinked its big eyes”}.

---

> > ### Author Response · Authors · 2024-11-20
> >
> > **Restrictive data.** If we understand the reviewer's comment correctly, the intervention features we consider in the experiments are indeed relatively simple, and we choose them because they exist in the feature spaces of Logit Lens, Tuned Lens, and the trained and labeled SAEs used and are also easily learned by probes and steering vectors. We believe these features to be interesting and insightful because they provide a “lower bound” of sorts on intervention. More specifically, if an explanation method doesn’t admit intervention on lower-level features, we cannot expect it to work on more safety-critical or high-level features we might care about in downstream tasks. Despite the simplicity of these features, the interpretability methods we evaluated were not able to intervene on model outputs without resulting in significant decreases to model coherence. We further present results on more complex features in Section B.3 of the Rebuttal Supplementary Materials in the Appendix.
> >
> > **Coherence definition.** We take coherence to mean the literal coherence of the generated text, as given in lines 261-268, where we note that it can be measured by any oracle (such as a human or high-performing LLM), and that we use Llama3.1-8b due to computational constraints with the prompt provided in lines 266-268. As suggested by reviewer 1BCv, we additionally consider using a rules-based grammar checker with the library LanguageTool to measure coherence via the number of grammatical errors in the generated text, and we find that the results remain consistent when using an LLM or the grammar checker. We will be sure to expand on the definition of coherence in the final paper.
> >
> > **Encoder-decoder vs mediation analysis.** The main goal of the encoder-decoder framework is to leverage existing interpretability methods that may not have been built for intervention and steering. In particular, many popular modern interpretability methods are correlational in nature (as noted by Mueller et al. (2024) and Saphra et al. (2024)), and may not explicitly provide mechanisms for intervention via the interpretations, such as Logit Lens or SAEs. Thus, the encoder-decoder framework allows us to evaluate methods for causal intervention, even if they were only built for interpretation or prediction instead of control, by decoding explanations such that we can measure their indirect effect. We do not think of this framework as “buying” anything over causal mediation analysis, but rather working towards an empirical and standardized evaluation of popular feature-based interpretability methods for the types of interventions discussed in the provided concurrent survey paper (which is noted as a suggestion for future work in Section 7.2.4).
> >
> > **Goal of the paper.** We reiterate that the goal of our paper is to evaluate interpretability methods that provide explanations of model representations on their ability to intervene on and steer model outputs - thus ensuring that interpretability can be used for control. To do so, we propose a unifying framework for four popular feature-based interpretability methods and metrics for successful steering of model outputs, allowing us to compare methods with different feature spaces. This framework is not the main goal of our work but simply allows us to use interpretability for intervention. We hope that this work encourages standardized evaluation of future interpretability methods and the development of explanation methods that permit intervention without degrading model performance and coherence.
> >
> > Mueller, Aaron, et al. "The quest for the right mediator: A history, survey, and theoretical grounding of causal interpretability." arXiv preprint arXiv:2408.01416 (2024).
> >
> > Saphra, Naomi, and Sarah Wiegreffe. "Mechanistic?." EMNLP 2024 BlackBoxNLP workshop (2024).

---

> > > ### Author Response · Authors · 2024-11-25
> > >
> > > Thank you again to the reviewer for your feedback on our work. Please let us know if we have addressed your comments, or if you have any remaining questions. We also kindly refer the reviewer to our discussion with Reviewer 1BCv, which further covers your points on causal mediation analysis and the benefits of the encoder-decoder framework. Thank you!

---

### Official Review · Reviewer_QT3J · 2024-11-04

**Soundness:** 3
**Presentation:** 3
**Contribution:** 3
**Rating:** 6
**Confidence:** 4

**Summary:**

This paper addresses the disconnect between interpretability and control in LLMs, proposing intervention as a core goal of interpretability. The authors introduce two novel metrics, intervention success rate and coherence-intervention tradeoff. They are designed to evaluate how well interpretability methods can effectively alter LLM outputs. The paper unifies four interpretability methods under an encoder-decoder framework, enabling human-interpretable feature interventions. Evaluated across three models (GPT-2, Gemma2-2b, Llama2-7b), the authors compare the framework against simpler methods like steering vectors and prompting. The study reveals limitations in existing methods, highlighting lens-based approaches as superior for simple interventions but indicating potential issues with performance degradation.

**Strengths:**

- The encoder-decoder framework consolidates multiple interpretability methods, allowing for unified evaluation and intervention capabilities
- The intervention success rate and coherence-intervention tradeoff metrics are intuitive ways of assessing the effectiveness of interventions and their impact on model coherence
- The paper offers insights into method-specific trade-offs
- Experiments are well-detailed, covering various aspects of intervention efficacy and model coherence, providing a comprehensive assessment across multiple models and features

**Weaknesses:**

- Lack of experiments on latest and advanced larger models. Llama-2 is not as advanced as the Llama3 to 3.2 family models, Gemma 2b is on the smaller side and GPT2 is quite dated. It would be interesting to see how this worked on some of the SOTA 7B+ models

- Some of the results in table 1 are missing, I couldn't find a reason for this, there could be a valid reason for this but having it in the table caption would be much clearer

- The evaluation primarily focuses on low-level features (e.g., “yoga,” “coffee”), which doesn't offer concrete insights into the framework’s utility for high-level or abstract features. It would be very interesting to see the set of intervention topics include more complex concepts. The authors do note that low-level features are more valuable, which I agree with, testing high level features would still be useful

- The coherence-evaluation setup primarily measures internal coherence without explicitly assessing task-related performance, which may differ in applied settings

**Questions:**

- As LLMs are intervened, edited, or steered with guardrails, behavior changes and concept drift may affect feature mapping and intervention fidelity. Does the framework account for this, or would it require retraining to maintain accurate interventions over time?

- How does the framework scale, e.g. for a 70b-200b model? What are the time and memory complexities?

- Is using Llama3 8b sufficient for coherence? It would be good to get some discussion on the limitations of this and if there are potential alternatives, e.g. using a human teacher etc

- Can the font and line width in the figures be made a little larger?

- In Appendix A.5 there is a broken reference

---

> ### Author Response · Authors · 2024-11-20
>
> Thank you for your suggestions! We appreciate your feedback on how to improve this work and address your concerns below.
>
> **Lack of experiments on latest/larger models.** We agree that the models we experiment on are relatively small; however, we were bound by the availability of trained, labeled SAEs and trained Tuned Lenses. Recently, SAELens released a trained and labeled SAE for a single layer of Llama3-8b, which we provide results for in Section B.1 of the Appendix, Rebuttal Supplementary Materials. Results on Llama3-8b are very similar to those for the models we included in the main paper, with Logit Lens and Tuned Lens outperforming the other intervention and interpretability methods, but underperforming prompting. Unfortunately, no trained SAEs/Tuned Lenses exist for larger variants (e.g. 70b-200b). We note that our framework should easily apply to larger models, and we have provided code that can be used for any models, lenses, and SAEs that may be released in the future.
>
> **Table 1 results missing.** We apologize for the confusion. The results for Gemma2-2b Tuned Lens and Llama2-7b SAE are missing because there are no publicly available trained SAEs/Tuned Lenses for those two models. Furthermore, as noted in lines 349-352, we cannot perform the reconstruction sanity check with steering vectors and probes as they do not generate complete explanations and thus cannot be inverted back to the original latent vector. We will be sure to reiterate this in the table caption.
>
> **Low- vs high-level features.** While we believe that intervention and control over high-level features is the end goal of many interpretability methods, we think that being able to intervene on simple, low-level features is a prerequisite to intervening on complex, high-level features. We also note that these are not necessarily distinct: we find anecdotally that upon steering the model toward a low-level feature (e.g. the word “yoga”), the topic of conversation also changes accordingly toward the underlying high-level concepts it represents.
>
> Furthermore, we note that we are limited in our ability to pick arbitrary or interesting features by the interpretability methods evaluated, in particular, because SAEs and lens-based methods do not allow for the specification of features. However, we present additional results for the features (1) responding in the French language (with English prompts) and (2) responding by yelling (writing in upper-case text) in Appendix Section B.3, Rebuttal Supplementary Materials. We find that, as expected, lens-based methods do not scale well to abstract features, and that steering vectors and probes are more performant than SAEs. However, overall, we see that the relationship between intervenability and coherence remains suboptimal, aligning with the existing results and conclusions of the paper.
>
> **Task-related performance.** Given that LLMs are used for a variety of downstream tasks but are generally built/tuned for chat performance, we believe that coherence is the best metric to maintain task-agnosticity, ensuring that the model retains its language modeling capability (i.e. its ability to generate coherent text without grammatical errors) after intervention. While we agree that using our framework to have fine-grained evaluation on various specialized downstream tasks is an interesting problem, we believe it to be beyond the scope of this paper.
>
> **Concept drift.** If we correctly interpret your question to mean asking whether the mapping from the model’s latent space to the interpretable latent space will be affected by the concept drift resulting from fine-tuning/editing with guardrails, then it depends on the explanation method used within the framework. For instance, because Logit Lens only uses the unembedding weight of the language model, no retraining would be required to maintain accurate interventions, as the unembedding matrix would have been updated during the model’s finetuning itself. However, probes and SAEs would have to be trained on the activations of the finetuned LM to still be valid. Thus, this limitation is not one of our framework, but of the underlying interpretability methods that may require additional data.

---

> > ### Author Response · Authors · 2024-11-20
> >
> > **Scalability/time and memory complexities.** This framework imposes nearly no additional computational cost during inference, depending on the explanation method used. In particular, for Logit and Tuned Lens, only two additional matrix multiplications and one in place setting operation are applied during the forward pass through the language model, and only the inverted unembedding matrix must be stored in memory, which is dependent on the size of the LLM used, but generally quite small relative to the full model. For SAEs, the time taken to use the SAE's encoder and decoder are added, and the SAE must be stored in memory (again, often much smaller than the size of the LM itself). Steering vectors and probes only require one addition operation and store one extra latent vector. For example, generating 30 tokens with Llama3-8b takes ~0.90 seconds, generating 30 tokens with intervention via Logit Lens takes ~0.95 seconds, generating 30 tokens with intervention via an SAE takes ~1.00 seconds, and generating 30 tokens with intervention via steering vectors or probes takes ~0.90 seconds. Should additional methods be included in this framework, the time and memory complexities would be fully dependent on those methods.
> >
> > **Coherence with Llama 3.1-8b.** Due to computational and budget constraints, we used the highest-performing open-source model we could to evaluate coherence. We agree that there are more accurate alternatives (such as human raters or a model finetuned on the task), but these options are either computationally- or time-intensive. In order to validate using Llama3.1-8b for measuring coherence, we perform the experiment suggested by reviewer 1BCv of “using rules-based tools to check for grammatical errors,” where we use a rules-based grammar checker (LanguageTool) to check for errors. Results are provided in Section B.2 of the Appendix, Rebuttal Supplementary Materials (page 18). We find that coherence is intuitively highly negatively correlated with the number of errors and that the results and conclusions remain consistent across both metrics. We will be sure to include more discussion of the limitations of an LLM-based coherence evaluator in the final paper.
> >
> > **Font width and broken reference.** Thank you for pointing these out! We will correct all typos and make the figures more legible in the final version of the paper.

---

> > ### Comment · Reviewer_QT3J · 2024-11-25
> >
> > I thank the authors for the detailed responses and for clarifying my misconception about the concept drift.
> >
> > With these updates I will be raising my score to 6.

---

### Author Response · Authors · 2024-11-20
**General Rebuttal**

We sincerely thank the reviewers for their thorough assessment of our paper and the AC for facilitating the discussion of our work. We appreciate the reviewers’ recognition that our paper is a “significant contribution to the field” (oPk4), with a “problem area that is well-motivated and timely” (1BCv), a method that is “promising” (LqMe), and “well-detailed” and “comprehensive” experiments (QT3J).
In the following section, we highlight the main contributions of our paper, summarize the main points made by reviewers, and respond to their comments. We also present additional experimental results, including results on (A) two more complex features: responding in the French language and yelling, (B) a larger and more recent language model, and (C) an alternative measure for coherence that uses a rules-based grammar checker as opposed to an LLM. Further details on the experiments and results are provided in Section B of the Appendix (titled Rebuttal Supplementary Materials, pages 18-19).

**Our main contributions.**
In this paper, we evaluate interpretability methods that aim to explain model representations with human-interpretable features on their ability to intervene on model representations and control outputs. More specifically, we argue that interpretability methods that aren’t built with causal interventions in mind should still be evaluated for their ability to control models, given that control and intervention are frequently the underlying goal for interpretability in the first place. In order to perform intervention with methods that were built only to predict feature presence (such as probes or Logit Lens) as well as fairly compare methods with different feature spaces, we unify four popular interpretability methods (Sparse Autoencoders, Logit Lens, Tuned Lens, and Probes) via an encoder-decoder framework that allows us to intervene on each explanation in the same manner. Please note that without the unification (and the proposed inverse explanation maps), it is not possible to intervene using these methods. We further propose (1) various success metrics for intervention and (2) introduce a prompt dataset to facilitate evaluation. Our main finding is that interpretability methods are far from optimal for intervention, and often underperform simpler alternatives like prompting and steering vectors.

**Relation to causal mediation analysis.**
Reviewers commented on the relationship between our work and past work on causal mediation analysis, which looks to build causal abstractions of large models to characterize the causal effect of latent representations. We highlight that our paper focuses on leveraging methods for **explaining** latent representations through human-interpretable feature predictions for the purpose of intervening on or controlling model outputs, which is the opposite setting to many causal mediation works, which instead try to understand models and circuits through causal analysis. Our main contribution is to present standardized evaluations of popular feature-based interpretability methods on their ability to intervene, which has been noted as an open problem and direction for future work in recent causal mediation papers such as those mentioned by reviewer LqMe.

**Low- vs high-level features.**
Reviewers commented on the generalizability of our results to more complex or real-world features. We note that we are limited by the feature spaces of the interpretability methods evaluated. In particular, we cannot specify the features learned by SAEs or encoded by Logit Lens or Tuned Lens. Thus, we can only evaluate them for the features they provide us, which are generally low-level token- or word-related features. That said, we include additional results for two more complex concepts: (1) responding in the French language and (2) responding by “yelling,” both of which are concepts that are not tied to specific tokens (see Section B.3 of the Appendix, Rebuttal Supplementary Materials). While results for safety-critical features would be interesting, we were unable to find such features that permitted steering in labeled and publicly available SAEs, leaving it an interesting topic for future work. We find that the results of our additional experiments align strongly with the conclusions drawn in the main paper.

**Definition and validation of coherence.**
Reviewers note that the definition of coherence is not very precise due to its reliance on an oracle, where human evaluation is difficult to obtain and LLM evaluation may be misaligned. Thus, as suggested by Reviewer 1BCv, we validate the coherence evaluations with a rules-based grammar checker (via the python library LanguageTool) to see if the number of grammatical errors is correctly negatively correlated with the LLM’s coherence scores. We find that results between coherence measured by an LLM and the grammar checker are very similar, with the same trends and conclusions drawn from both.

---

### Meta-Review · Area_Chair_2e9n · 2024-12-24

**Metareview:**

This paper presents a framework for evaluating the effectiveness of interpretability methods in large language models (LLMs) through the lens of "intervention," aiming to bridge the gap between interpretability and model control. The authors introduce two novel evaluation metrics: intervention success rate and coherence-intervention tradeoff, designed to quantify how well interpretability methods alter LLM outputs. The proposed framework consolidates four distinct interpretability methods (SAEs, logit/tuned lens, probing, and lens-based approaches) under an encoder-decoder architecture. It evaluates their performance across multiple LLMs (GPT-2, Gemma2-2b, and Llama2-7b). The paper compares the framework to simpler methods such as steering vectors and prompting, uncovering performance trade-offs and highlighting the advantages and limitations of each approach.

The methodology is largely sound, and the core idea of evaluating interpretability methods through intervention-based metrics is promising.  Despite these merits of this paper, several weaknesses remain that were only partially addressed in the rebuttal. First, reviewers highlighted the limited experimentation with more complex or real-world features. Although the authors incorporated experiments on two more complex concepts (responding in French and yelling), they are still constrained by the feature spaces of the interpretability methods they evaluated. The complexity of features, such as safety-critical ones, remains an important area for future work. Furthermore, the authors' reliance on low-level features like tokens and words may limit the generalizability of their findings to higher-level, more abstract concepts, a concern that was raised by the reviewers. The rebuttal does not fully resolve these issues, and a broader exploration of feature complexity would strengthen the paper's conclusions.

Second, the definition and validation of "coherence" were criticized for their lack of precision and reliance on an oracle, which complicates human evaluation and may misalign with LLM-based assessment. Although the authors attempted to address this by validating coherence with a rules-based grammar checker, the results still depend heavily on the specific tools used, and the fundamental issues with LLM-based evaluation persist. The authors' approach to coherence remains somewhat imprecise and lacks rigor, suggesting a need for more robust validation techniques. In addition, while the paper distinguishes itself from causal mediation analysis, it is worth noting that causal mediation could provide a more structured framework for understanding the causal impact of interpretability methods on model outputs. A clearer discussion of how their intervention-based approach contrasts with or could benefit from causal analysis would strengthen the theoretical foundation of the paper. These unresolved issues point to the need for more comprehensive experimental design and methodological refinement, particularly in the context of causal relationships and the evaluation of more complex, high-level features.

Although this submission at the current stage has some merits, more efforts are still needed before it can be accepted.

**Additional Comments On Reviewer Discussion:**

During the rebuttal period, several key points were raised by the reviewers, and the authors made efforts to address these concerns, though some issues remained only partially resolved.

Complexity of Features and Generalizability: Reviewers noted the paper's limited scope in evaluating more complex or real-world features, with a heavy emphasis on low-level token-based features. The authors responded by adding experiments involving more complex concepts, such as responding in French and yelling. However, these features, while more complex than tokens, still did not fully address the need for testing with higher-level, safety-critical features. The authors acknowledged the limitations of the methods' feature spaces, particularly with respect to SAE, Logit Lens, and Tuned Lens, and highlighted the need for future work in this area. Despite the authors’ efforts, the complexity of the features remained a concern, and it was noted that further exploration of real-world, high-level features is essential for the paper's broader applicability.

Coherence Definition and Validation: Reviewers expressed concerns over the lack of clarity in the definition of "coherence," particularly because the original approach relied on an oracle for evaluation. The authors responded by validating the coherence measure using a rules-based grammar checker, which showed results consistent with the original LLM-based evaluation. However, the reliance on external tools, such as the grammar checker, left some reviewers unconvinced regarding the precision and rigor of the coherence evaluation. The rebuttal provided some improvements, but the underlying issues with LLM-based evaluation methods and their alignment with human evaluation were not fully resolved. The reviewers still questioned whether the coherence metric was sufficiently rigorous and generalizable.

Causal Mediation and Interventions: Reviewers questioned the relationship between the paper’s contributions and causal mediation analysis, specifically regarding how interpretability methods could be used for causal interventions. The authors clarified that their focus was not on building causal abstractions but on evaluating interpretability methods for their ability to intervene and control model outputs. While this distinction was addressed, the reviewers felt that the paper could benefit from a clearer connection to causal mediation analysis, particularly in terms of leveraging causal relationships for intervention. The authors provided some clarifications but did not significantly deepen the connection to causal mediation techniques, leaving this point somewhat unresolved.

In my final decision, I weighed these points carefully. While the authors made some improvements, particularly in the addition of new experiments and the refinement of the coherence validation process, several important issues remained unresolved. The limited scope of feature complexity and the concerns regarding the coherence evaluation methodology indicated that further refinement was needed. Additionally, the paper’s relationship to causal mediation analysis could have been better developed. While the authors' contributions remain valuable, especially in advancing interpretability for model intervention, the remaining gaps suggest that further work is required to fully address the reviewers' concerns.

---

### Decision · Program_Chairs · 2025-01-22

Reject